# Cellular Senescence in Hepatocellular Carcinoma: Immune Microenvironment Insights via Machine Learning and In Vitro Experiments

**DOI:** 10.3390/ijms26020773

**Published:** 2025-01-17

**Authors:** Xinhe Lu, Yuhang Luo, Yun Huang, Zhiqiang Zhu, Hongyan Yin, Shunqing Xu

**Affiliations:** 1School of Life and Health Sciences, Hainan University, Haikou 570228, China; 2School of Public Health, Tongji Medical College, Huazhong University of Science and Technology, Wuhan 430030, China; 3School of Environmental Science and Engineering, Hainan University, Haikou 570228, China; 4School of Tropical Agriculture and Forestry, Hainan University, Haikou 570228, China

**Keywords:** hepatocellular carcinoma, cellular senescence, machine learning, immune microenvironment, NK cells, *BIRC5*

## Abstract

Hepatocellular carcinoma (HCC), a leading liver tumor globally, is influenced by diverse risk factors. Cellular senescence, marked by permanent cell cycle arrest, plays a crucial role in cancer biology, but its markers and roles in the HCC immune microenvironment remain unclear. Three machine learning methods, namely k nearest neighbor (KNN), support vector machine (SVM), and random forest (RF), are utilized to identify eight key HCC cell senescence markers (HCC-CSMs). Consensus clustering revealed molecular subtypes. The single-cell analysis explored the tumor microenvironment, immune checkpoints, and immunotherapy responses. In vitro, RNA interference mediated *BIRC5* knockdown, and co-culture experiments assessed its impact. Cellular senescence-related genes predicted HCC survival information better than differential expression genes (DEGs). Eight key HCC-CSMs were identified, which revealed two distinct clusters with different clinical characteristics and mutation patterns. By single-cell RNA-seq data, we investigated the immunological microenvironment and observed that increasing immune cells allow hepatocytes to regain population dominance. This phenomenon may be associated with the HCC-CSMs identified in our study. By combining bulk RNA sequencing and single-cell RNA sequencing data, we identified the key gene *BIRC5* and the natural killer (NK) cells that express *BIRC5* at the highest levels. *BIRC5* knockdown increased NK cell proliferation but reduced function, potentially aiding tumor survival. These findings provide insights into senescence-driven HCC progression and potential therapeutic targets.

## 1. Introduction

Hepatocellular carcinoma (HCC) is a significant form of liver cancer, representing the most common primary liver tumor globally [1]. Various risk factors are reported to be associated with HCC, including chronic hepatitis B and C infections, liver cirrhosis, metabolic factors, and genetic alterations [2,3,4]. Early-stage HCC often has no symptoms, resulting in delayed diagnosis and tumor progression. Additionally, effective treatments are scarce for advanced HCC [5]. The molecular pathogenesis of HCC involves complex mechanisms such as allelic losses, chromosomal changes, gene mutations, and alterations in cellular pathways [6,7]. Despite the growing investigations for therapeutic guidance, the prognosis prediction for HCC patients remains unsatisfactory. Therefore, innovative biomarkers are urgently needed, as well as a clearer understanding of HCC for developing targeted therapies and improving patient outcomes.

Cellular senescence, a stable cell cycle arrest, can be triggered by factors such as mitotic stress, carcinogenic activation, tissue damage, telomere shortening and structural changes, oxidative and genotoxic stress, ionizing radiation, epigenetic and chromatin alterations, protein disturbances, mitochondrial dysfunction, inflammation, radiation therapy, or chemotherapy [8,9]. These years, accumulating evidence shows that both aging and cancer share a set of hallmarks [10]. Certain aging hallmarks, including genomic instability, epigenetic changes, chronic inflammation, and dysbiosis, promote cancer, while others like telomere shortening and stem cell depletion may inhibit its development [11,12,13,14]. Therefore, understanding the underlying cellular senescence in cancer is crucial for developing targeted therapies and improving patient outcomes.

Cellular senescence in HCC affects tumor growth, therapy response, metabolism, immune function, and the tumor microenvironment. Acting as a double-edged sword, senescence can inhibit tumor progression by inducing cell cycle arrest and activating immune surveillance, yet it can also promote tumor growth and therapy resistance through the senescence-associated secretory phenotype (SASP) that alters the surrounding environment [15,16,17]. Senescent cells interact with the immune system by attracting T cells and macrophages to boost anti-tumor immunity, but the SASP can also foster an immunosuppressive environment that aids tumor progression and metastasis [18,19]. Moreover, metabolic reprogramming in senescent cells can influence the tumor microenvironment, affecting how immune cells function and respond to therapy [20]. On HCC, it has been highlighted that cellular senescence is closely linked to the occurrence, development, and therapy of tumors [18,21]. Studies have also demonstrated the impact of cellular senescence on energy metabolism, immune infiltration, and response to immunotherapy in HCC, highlighting the multifaceted role of senescence in shaping the tumor microenvironment [22,23]. Additionally, cellular senescence in HCC is characterized by multiple markers, such as morphological changes, expression of cell cycle inhibitors, and alterations in the nuclear membrane, emphasizing the complexity of senescence in the context of liver cancer [24]. Despite substantial research, the markers and roles of cellular senescence-related genes in HCC remain unclear. We believe that further investigation of these genes could enhance prognostic predictions, deepen our understanding of tumor biology, and improve therapeutic strategies for this aggressive cancer. Understanding these dynamics will be essential for developing effective strategies to target senescence in HCC therapy.

In this study, we used various machine learning algorithms to identify HCC cell senescence markers (HCC-CSMs) and analyzed two distinct clusters based on these markers, correlating them with clinical characteristics and mutation patterns. Using single-cell RNA-seq data, we found that increased immune cell numbers allow hepatocytes to reclaim more of the cell population, likely linked to the identified HCC-CSMs. Additionally, we observed that immune evasion intensifies as cancer progresses by examining immune checkpoint gene expression and immune dysfunction. In early disease stages, BIRC5 expression was higher in natural killer/T (NK/T) cells than in hepatocytes, but it significantly decreased as HCC advanced, mirroring changes in NK/T cell proportions. Consequently, we focused on BIRC5’s role in NK cells. Our in vitro experiments showed that knocking down BIRC5 increased NK cell numbers and proliferation but reduced their immune function and cytotoxicity, potentially facilitating cancer cell growth. Collectively, our computational and experimental analyses reveal how cellular senescence in the immune microenvironment drives HCC progression, and they identify potential therapeutic targets for liver cancer treatment.

## 2. Results

### 2.1. Stronger Predictive Abilities of Cellular Senescence-Related Genes on HCC Development than Differential Expression Genes (DEGs)

Cancer development and the selection of markers are typically determined by identifying differentially expressed genes obtained from comparing cancer samples with normal ones [25,26]. Considering the increasing commonalities between cancer and aging [10,27], we assume that senescence-related genes may provide better predictions of cancer development. To explore this, we compared the predictive abilities of DEGs (Appendix A) and senescence-related genes on HCC. The senescence-related genes (the CellAge gene set; Appendix A) were obtained from the Human Ageing Genomic Resources database. Meanwhile, the liver cancer samples of the TCGA-LIHC dataset and the ICGC-LIRI dataset were employed as the training cohort and testing cohort, respectively, with the clinical outcomes provided. Subsequently, least absolute shrinkage and selection operator (LASSO) regression analysis was used on the training cohort to delineate potential associations (Figure 1A). In the LASSO regression, the lambda value represents the size of the regularization parameter. Adjusting the lambda value controls the model’s sparsity, with larger values increasing regularization penalties and smaller values making it closer to ordinary least squares regression. We used “partial likelihood deviance” to evaluate the model’s predictive ability, where lower values indicate a better fit (Figure 1B). Different lambda values were applied to different gene sets in the regression analysis. To evaluate the trained LASSO model, predictions were made on the testing cohort, and receiver operating characteristic (ROC) curves were plotted, with area under the curve (AUC) calculated for different survival times. Interestingly, the results showed that AUCs of the model based on senescence-related genes were larger than those of DEGs, especially for 5-year survival (senescence-related genes: 5-year AUC = 0.849; DEGs: 5-year AUC = 0.545), indicating the stronger predictive ability of senescence-related genes than commonly used DEGs, particularly in long-term survival prediction (Figure 1C). Consistently, patients in the ICGC-LIRI dataset were divided into low-risk and high-risk groups based on risk score calculations. Kaplan–Meier (K-M) survival analysis was conducted to assess the prognosis of patients in different groups. We calculated the risk score based on the genes selected by LASSO and classified patients with a risk score higher than the median as the high-risk group and the rest as the low-risk group. The Kaplan–Meier plot showed that the prognostic significance between the two groups using genes selected by CellAge (*p* = 0.00011) was more significant than that of genes selected from the DEG analysis (*p* = 0.019). This suggests that the aging-related gene set has predictive power for long-term survival in HCC (Figure 1D).

### 2.2. Identification of HCC-CSMs Using Multiple Machine Learning Algorithms

Based on the previous LASSO survival regression analysis, we attempted to obtain a list of non-zero regression coefficients representing genes for identifying cancer samples. However, when we used the obtained 10 genes (CBS, CD34, EID3, ENO1, IGFBP1, INPP4B, PON1, SERPINE1, WEE1, and YBX1) as marker genes for HCC discrimination, the results showed that although the combined regression analysis of these senescence-related genes could predict survival time with high accuracy, the majority of genes alone were not effective in discriminating cancer samples (Appendix A). We divided patients into a high-expression group (exp_high) and a low-expression group (exp_low) based on the median expression levels of different genes and compared the prognostic differences between the two groups. Among them, only ENO1, WEE1, and YBX1 showed significant differences (*p* < 0.05), while CBS, CD34, EID3, IGFBP3, INPP4B, PON1, and SERPINE1 showed no significant differences (*p* > 0.05). In the age of big data, machine learning plays a crucial role in analyzing large datasets to derive valuable insights and make accurate predictions [28]. To further acquire cell senescence markers associated with HCC, we aimed to employ typical supervised machine learning algorithms, including k nearest neighbor (KNN), random forest (RF), and support vector machine (SVM), to identify pivotal features linked with HCC. Each machining learning method is applied to the labeled training data (including 50 normal and 371 HCC individuals from TCGA-LIHC) (Figure 2A). We used recursive feature elimination to rank the feature genes through 10-fold cross-validation. The number of features was selected based on the mean squared error (MSE), root mean squared error (RMSE), and Rsquared for different feature sets, where lower MSE and RMSE values and higher Rsquared values are better. By considering all three parameters, we selected 27, 13, and 16 candidate HCC-related cell senescence markers, as shown in the Venn diagram (Figure 2C). Collectively, we obtained eight overlapped markers from different techniques and defined them as the HCC-CSMs, including NOX4, BIRC5, E2F1, CD34, KIF2C, AURKA, CDK1, and GMNN (Appendix A). The biomarkers of HCC mentioned in previous studies include BIRC5 [29], E2F1 [30], CD34 [31], KIF2C [32], AURKA [33], CDK1 [34], and GMNN [35]. However, there has been no research identifying NOX4 as a biomarker for HCC so far. Furthermore, to evaluate the classification power of individual genes, predictions were made on the testing labeled cohort (from ICGC-LIRI, N = 437), and ROC curves were plotted, with the AUC calculated. Remarkably, the encouraging result showed that all eight genes exhibited AUC values surpassing 0.85, underscoring their efficacy in discriminating between HCC and normal liver tissues (Figure 2D). To further validate the reliability of HCC-CSMs, we also performed validation using data from GSE214846 (65 HCC samples and 65 normal samples), and the AUC values for all eight genes remained greater than 0.85 (Appendix A).

### 2.3. Examining the Clinical and Molecular Signatures of HCC Clusters Identified by HCC-CSMs

We have shown that HCC-CSMs can effectively distinguish normal liver tissue from HCC. However, we also want to determine whether HCC-CSMs can reflect the severity of HCC. To further explore the expression patterns of HCC-CSMs within HCC patients, we attempted to group the HCC patients from the TCGA-LIHC dataset, referring to the expression levels of eight HCC-CSMs. Therefore, we performed consensus clustering analysis using the expression data of HCC-CSM genes sourced from TCGA-LIHC. Determining the optimal number of clusters (k) involved analyzing the consensus cumulative distribution function (CDF). Our observation revealed that the CDF stabilized at k = 2, suggesting that two clusters were the most suitable representation (Figure 3A). Meanwhile, when utilizing t-distributed stochastic neighbor embedding (t-SNE) for dimensionality reduction and visualization of patients with the expression levels of HCC-CSMs, we observed a clear delineation into two clusters as well (Figure 3B).

Furthermore, to uncover the clinical variances among these two different clusters, K-M survival analysis was performed to evaluate the prognosis of patients in different clusters. It demonstrates that although both groups experienced a rapid decline in survival rates over time, there remains a significant difference in survival rates between the different clusters (Figure 3C), with the survival time of patients in cluster one experiencing a more rapid decline. This result indicates that the HCC-CSMs we obtained not only characterize the presence of cancer but also correlate with cancer progression. Based on this, to further explore the correlation of clinical characteristics of these two clusters, we utilized the meta-information of the patients and calculated the HR on individual clinical characteristics. We found that the different clusters had no significant relationship with age and gender, but cluster 1 exhibited a high correlation with cancer progression featured by grade, pathologic T, and stage (Figure 3D). For instance, there is a significant difference in the stage distribution between cluster 1 and cluster 2 (*p* = 0.000964), with cluster 1 showing a more advanced overall stage. Together, cluster 1 showed a markedly more severe degree of deterioration, suggesting that our clustering by identified HCC-CSMs can predict the progression of HCC.

Considering the significant differences in clinical phenotypes between cluster 1 and cluster 2, we aim to conduct a more in-depth analysis of the differentially expressed genes (Appendix A), particularly focusing on the expression patterns of the identified HCC-CSMs between these two groups. Notably, by bulk-RNA seq, genes such as *BIRC5*, *KIF2C*, *E2F1*, and *CDK1* manifested elevated expression levels within cluster 1, contrasting with *AURKA*, *GMNN*, *MOX4*, and *CD34*, which exhibited no significant differences between the two clusters (Figure 4A). Intriguingly, among these markers, referring to the cellular senescence profiles of CellAge, it is reported that inhibiting the expression of the *BIRC5*, *KIF2C*, and *CDK1* genes may promote cellular senescence [36,37,38]. *E2F1* was also induced in cluster 1, while limited evidence has shown that both overexpression and downregulation of *E2F1* are related to cellular senescence [39,40]. The differential expression of these significantly senescence-related markers suggests that cellular senescence is an important factor in disease development. In addition, we conducted Gene Ontology (GO) and Kyoto Encyclopedia of Genes and Genomes (KEGG) enrichment analyses to elucidate the functional implications of differentially expressed genes. GO analysis revealed enrichment in crucial processes such as nuclear chromosome segregation and cellular division (Figure 4B), shedding light on the molecular mechanisms underpinning HCC progression and cellular senescence. Similarly, KEGG analysis highlighted pathways primarily associated with cell division processes (Figure 4C), emphasizing their integral role in regulating cellular proliferation and organization, thereby contributing to HCC development and cellular senescence. The gene enrichment analysis of HCC-CSMs revealed its involvement in biological processes such as cellular division and the regulation of cellular proliferation and organization (Figure 4D).

In our previous analysis, we identified two distinct clusters that exhibit a strong correlation with HCC staging. The relationship between somatic mutations and HCC staging is a critical area of research, as it provides valuable insights into tumor progression. Studies have shown that as HCC advances from early to late stages, somatic mutations—particularly those in key oncogenes and tumor suppressor genes—accumulate progressively. Notably, mutations in the *TP53* and *CTNNB1* genes are among the most frequent in HCC, and their incidence increases in the later stages of the disease [41,42]. Therefore, we explored the distribution of gene mutations between cluster 1 and cluster 2. The comprehensive landscape of somatic variations depicted the mutation patterns of the top 10 most frequently mutated driver genes (Figure 4E). Research indicates that *TP53* mutations, which play a critical role in cell cycle regulation and apoptosis, are present in approximately 18.7% of HCC cases, with higher frequencies observed in advanced stages [43]. The results revealed that gene mutations were predominantly concentrated in deletions and missense mutations. Notably, the *TP53* gene, responsible for encoding a tumor suppressor protein, exhibited a significantly higher number of gene mutations in cluster 1 compared to cluster 2. Additionally, in cluster 1, *TP53* gene mutations were mostly deletions, while in cluster 2, they were primarily missense mutations. Furthermore, the forest plot illustrated that the mutations in *TP53*, *TRB1*, *IRX1*, and *SPEG* genes were markedly more abundant in cluster 1 than in cluster 2 (Appendix A). The differences in patterns of gene mutations between cluster 1 and cluster 2 suggest distinct genetic variation characteristics among these clusters, reflecting differences in tumor progression mechanisms. These findings provide important clues for a deeper understanding of the genetic characteristics and tumor development mechanisms between different tumor clusters, which will help guide the formulation of personalized treatment strategies.

### 2.4. Exploration of Immune Microenvironment by HCC-CSMs

Recent evidence has documented that infiltrating distinct immune cell populations within the immune microenvironment influences tumor development differentially [27,44]. Considering the current trend where cancer-related markers are increasingly examined at the single-cell level [45,46], we assumed that the HCC-CSMs we have identified should exhibit heterogeneity across different cell types at the single-cell scale and tried to explore their roles in the microenvironment. Accordingly, single-cell transcriptome data from 10 primary tumors samples (method) were analyzed to assess the proportion of the different types of cells and the corresponding expression patterns of HCC-CSMs in various cell types.

The result (Figure 5A) reveals the stage-specific alterations in cell type composition during disease progression, with a marked increase in the relative abundance of NK/T cells at stage IIIA. In contrast, hepatocytes exhibit a notable decrease in abundance at stage IIIA, followed by a significant increase at stage IIIB and stage IV, potentially reflecting a compensatory response or altered hepatic microenvironment in advanced stages. These results highlight the dynamic and heterogeneous nature of the cellular microenvironment, emphasizing the critical roles of immune modulation and hepatocyte adaptation during disease progression. Interestingly, hepatocytes are able to “make a comeback” and regain a larger proportion of the cell population as NK cell numbers increase. Additionally, considering the HCC-CSMs are highly related to cell division, we assume that they contribute to this phenomenon. To address this issue, the proportion of cells expressing these markers at different stages was evaluated. Moreover, we examined the expression correlations among these genes and detected the highly correlated relationship between two genes (*CD34* and *NOX4*, Figure 5B) and the highly correlated relationship between the other six genes (Figure 5C). Furthermore, by illustrating the proportion of cells expressing HCC-CSMs at different stages (Figure 5D, Appendix A). the results show that the *NOX4* and *CD34* are primarily expressed in endothelial cells throughout the progression of HCC, while other genes, however, are mainly expressed in hepatocytes and NK/T cells. *CD34* acts as a marker to signify states such as tumor angiogenesis, which may provide nutrients and oxygen to tumors, thereby promoting their growth and dissemination. *NOX4*, acting as a type of NADPH oxidase responsible for generating reactive oxygen species (ROS) within cells, is associated with cellular oxidative stress, which is a key mechanism leading to cellular senescence. Additionally, it may modulate the microenvironment, thereby promoting the onset and progression of cancer, indicating that a high expression of these two cellular senescence markers, *CD34* and *NOX4*, in endothelial cells, could be potential markers for tumor development.

For other genes, the proportion changes in different cell types expressing *BIRC5*, *CDK1*, *AURKA*, *E2F1*, and *KIF2C* are remarkably similar to the overall proportion changes in different cell types during cancer progression. Among these genes, *BIRC5* plays a crucial role in various human malignancies by regulating cell proliferation, apoptosis, and resistance to genotoxic agents [47]. *KIF2C*, *CDK1*, and *AURKA* play crucial roles in cell division or act as cell cycle regulatory genes [48,49,50,51]. E2F1, a member of the E2F family of transcription factors, is crucial in regulating cell cycle progression, apoptosis, and DNA repair processes in human cells [52,53]. Considering consistent changes in the whole cell proportions (Figure 5A) and the cell proportions expressing some markers (Figure 5D), we are curious about the expression patterns in the individual markers in each cell type. Accordingly, we compared the average expression levels of different markers (Appendix A) in divergent types of cells along the progression. Most intriguingly, *BIRC5*, which exhibits the highly differently expressed genes along the progression by RNA-seq (Figure 4A), shows a higher average expression in NK/T cells than hepatocytes in the early stages (Figure 5E). However, as the disease progresses, particularly during the critical stage IIIA, *BIRC5* expression is significantly downregulated. In the later stages, the average expression of *BIRC5* in NK/T cells falls below that in hepatocytes. This intriguing result directly underscores the importance of markers associated with cell division or senescence in disease progression. Additionally, considering the varying proportions of cell types at different stages of disease progression, this finding may also explain why we identified higher *BIRC5* expression in cluster 1 (Figure 4A). We conduct a series of in vitro experiments to validate this result further, which will be elaborated on in detail later in the text. Overall, we observed that most of these cell senescence markers are associated with cell division or the cell cycle, highlighting the importance of cell division in hepatocytes and NK/T cells during disease progression. These genes may play crucial roles in driving changes in the immune microenvironment at the single-cell level.

At the single-cell level, we particularly focused on the changes in NK and T cell populations across different stages of HCC progression. We performed functional enrichment analysis on the differentially expressed gene sets from these immune cell types. Based on our hypothesis, while NK cell numbers increase in later stages, their cytotoxicity does not improve. Our enrichment analysis reveals that the role of immune cells is gradually weakening over the progression of the disease. In the earlier stages, NK/T cells enriched pathways such as natural killer cell chemotaxis, the positive regulation of T cells, and T cell chemotaxis. However, in later stages, these pathways were not enriched, reflecting a decline in the immune cells’ functional mechanisms (Appendix A).

Based on the above analysis, considering the significant changes detected in the proportion of immune cells during the disease progression, we assume that during disease progression, there may be an increase in the number and proportion of immune cells, such as NK cells. However, despite this increase, these cells’ overall immune function or cytotoxicity may decline, thereby creating a more favorable environment for cancer cell survival. This could lead to an increase in cancer cell numbers and further disease progression. To further predict the sensitivity to immunotherapy along the progression, we compared the expression levels of immune checkpoint genes in cluster 1 and cluster 2. Notably, the most commonly targeted checkpoints for cancer immunotherapy, such as *PDCD1*, *CTLA4*, *LAG3*, and *TIGHT* exhibited a significantly higher expression in cluster 1 compared to cluster 2 (Figure 5F). A high expression of immune checkpoint genes may be associated with the degree of immune escape in the tumor microenvironment, indicating that tumors in cluster 1 may have a higher capability for immune escape, potentially resulting in low sensitivity or resistance to immunotherapy. Conversely, a lower expression of immune checkpoint genes in cluster 2 may indicate that this tumor type is more suitable for immunotherapy, as the inhibition of immune checkpoints may more effectively activate immune responses. Furthermore, we computed the tumor immune dysfunction and exclusion (TIDE) score to predict the response of tumors to immune therapy (Figure 5G). Consistently, the result revealed that cluster 1 exhibited a higher TIDE score than cluster 2, indicating a lower likelihood of response to immune therapy for the tumor. Furthermore, we observed a lower immune dysfunction score and a higher exclusion score in cluster 1, suggesting that compared to cluster 2, immune exclusion in this cluster may compromise the intended effects of immunotherapy. This indicates the potential differences in immune therapy response among different clusters and provides valuable information for determining immune therapy strategies. More interestingly, based on our analysis, we found that the characteristics of cluster 1 indicate a more advanced stage of disease progression. Consequently, this result suggests that the affected individuals exhibit stronger immune evasion features as the disease worsens, leading to a diminished response to immunotherapy.

### 2.5. In Vitro Validation of BIRC5’s Role in Modulating NK Cell Proliferation and Cytotoxicity

To gain insights into the role of NK/T cells in cancer and their involvement in tumor immune evasion and to develop new immunotherapy strategies, we conducted further experimental validation based on the hypothesis regarding HCC-CSMs in cancer progression. In the transcriptomic analysis, we observed that the expression of *BIRC5* was the most significantly high in cluster 1 compared to cluster 2 (Figure 4A). Single-cell analysis further revealed that at the early stages of the disease, *BIRC5* expression in NK/T cells was higher than in hepatocytes. However, as the disease progressed, particularly during the critical IIIA phase, *BIRC5* expression was markedly downregulated. In the later stages, the average expression of *BIRC5* in NK/T cells was lower than in hepatocytes (Figure 5E), which paralleled the changes in the proportion of NK/T cells observed at different disease stages (Figure 5A). Therefore, we focused on *BIRC5* and its role in NK cells. Small interfering RNA (siRNA) knockdown of the *BIRC5* gene was performed in the NK92 cell line. After transfection with si-BIRC5-1, si-BIRC5-2, and si-BIRC5-3, the mRNA silencing efficiency was assessed by real-time RT-PCR. Compared to un-transfected cells (NK92) and the empty-vector control (NK92 + siRNA-NC), *BIRC5* expression was significantly downregulated in the si-BIRC5 groups, particularly in the si-BIRC5-2 group, which was selected as si-BIRC5 for subsequent experiments (Figure 6A). The CCK-8 assay results show that compared to the control group NK92 + siRNA-NC (NK92 cells transfected with siRNA-NC), the group NK92 + si-BIRC5 (NK92 cells with *BIRC5* knockdown via si-BIRC5) exhibited a significant increase in proliferation, indicating that *BIRC5* knockdown enhances NK92 cell proliferation (Figure 6B). Flow cytometry analysis of cell cycle distribution revealed a reduction in the G1 phase population and an increase in the G2 phase population after *BIRC5* knockdown, suggesting that *BIRC5* knockdown facilitates the G1 to G2 phase transition and may regulate cell cycle progression (Figure 6C). Additionally, flow cytometry analysis of ROS levels showed that the group NK92 + siRNA-NC had high ROS fluorescence intensity, indicating oxidative stress. In contrast, the group NK92 + si-BIRC5 had significantly lower ROS levels, suggesting that *BIRC5* knockdown reduces ROS production in NK92 cells (Figure 6D). Overall, these results demonstrate that *BIRC5* knockdown promotes NK92 cell proliferation, alters cell cycle distribution, and reduces oxidative stress, suggesting that *BIRC5* plays a key role in regulating NK92 cell division.

To further investigate the changes in cytotoxicity of NK cells against HCC cells after proliferation, we established a co-culture system and transwell system using NK92 cells and HepG2 cells (Figure 6E). In this system, NK92 cells were suspended in the upper chamber, while HepG2 cells were cultured in the lower chamber. The results of the CCK-8 assay showed that group HepG2 (NK92 + siRNA-NC) demonstrated a lower proliferation rate of HepG2 cells when co-cultured with NK92 cells transfected with siRNA-NC (Figure 6F). However, in group HepG2 (NK92 + si-BIRC5), where *BIRC5* was knocked down in NK92 cells, the HepG2 cell proliferation rate increased, suggesting that *BIRC5* knockdown in NK92 cells diminishes their cytotoxicity towards HepG2 cells, leading to enhanced HepG2 proliferation. Furthermore, the flow cytometry-based apoptosis assay results show the apoptosis rates of HepG2 cells under different conditions (Figure 6G). NK92 cells transfected with siRNA-NC significantly induced apoptosis in HepG2 cells. However, in group HepG2 (NK92 + si-BIRC5), where NK92 cells were transfected with si-BIRC5, the apoptosis rate of HepG2 cells dropped. This reduction in apoptosis suggests that *BIRC5* knockdown in NK92 cells impaired their ability to induce apoptosis in HepG2 cells. Furthermore, the transwell migration assay results reveal a significant difference in the migration ability of HepG2 cells co-cultured with NK92 cells under different conditions. In group HepG2 (NK92 + si-BIRC5), the migration count of HepG2 cells was substantially higher than group HepG2 (NK92 + siRNA-NC), suggesting the knockdown of *BIRC5* in NK92 cells possessing a reduced ability to suppress HepG2 cell migration (Figure 6H). The transwell invasion assay results illustrate that HepG2 cells in the *BIRC5* knockdown group HepG2 (NK92 + si-BIRC5) demonstrated a markedly higher invasion count, indicating the weakened ability to suppress the invasion of HepG2 cells (Figure 6I). To test whether siRNA acts directly on HepG2 cells, we have conducted qPCR analysis to confirm the expression of *BIRC5* in HepG2 cells after the co-culture. The results showed no significant change in *BIRC5* expression in HepG2 cells (Appendix A). This confirms that the siRNA exclusively targeted NK92 cells and had no impact on HepG2 cells.

Overall, based on our detailed analysis in silico, we hypothesize that during disease progression, there may be an increase in both the number and proportion of immune cells, such as NK cells and T cells. However, despite this increase, the overall immune functionality or the cytotoxicity of these cells may decline, thereby creating a more favorable microenvironment for cancer cell survival. This, in turn, could lead to an increase in cancer cell proliferation and further disease progression. Our experimental results are consistent with this hypothesis. Specifically, we observed that following *BIRC5* knockdown, the proliferation of NK immune cells increased. However, despite this increase, the NK cells were unable to effectively inhibit the HCC cells and exhibited reduced cytotoxicity. This molecular experiment suggests that HCC cells may acquire a greater ability for immune evasion, aligning with our analysis of immune escape mechanisms in the information section. Moreover, computational analyses, supported by experimental data, revealed an increased expression of immune checkpoint genes in specific clusters (Figure 5F,G). This further implicates HCC-CSMs in facilitating immune escape and highlights their contribution to immune modulation and tumor progression in HCC.

Collectively, we believe this integration of computational and experimental approaches strengthens the reliability and significance of our findings. This supports the notion that immune escape plays a pivotal role in facilitating cancer progression, even in the presence of increased immune cell numbers.

## 3. Discussion

### 3.1. Cell Senescence Markers in HCC

In recent years, significant strides have been made in the utilization of machine learning methodologies for the identification of pivotal genes implicated in cancer etiology [54,55]. In our study, we employed multiple machine learning algorithms to incorporate senescence-related genes into predictive models for HCC, a novel approach that could enhance prognostication in HCC patients. Cellular senescence-related genes demonstrated stronger predictive abilities than DEGs, especially for long-term survival, underscoring the significance of senescence in cancer development and progression. Additionally, identifying HCC-CSMs effectively distinguished HCC from normal liver tissues and highlighted their critical role in cancer progression.

Among the eight HCC cell senescence markers (HCC-CSMs) identified, some significantly influence cancer progression while others remain understudied. *BIRC5* is crucial in HCC, with studies demonstrating that both knockout and overexpression significantly affect HCC cell survival [56]. However, the impact of *BIRC5* on NK cells remains unexplored. E2F1 regulates HCC progression through multiple pathways; Shen et al. showed that E2F1 activates KDM4A-AS1 via the PI3K/AKT pathway [57], while Lei et al. suggested that ARRB1 plays a critical role in HBV-related HCC by modulating autophagy and the CDKN1B-CDK2-CCNE1-E2F1 axis [58]. KIF2C, part of the kinesin superfamily, is implicated in various cancers, with overexpression observed in breast, lung, and bladder cancers [59,60,61]. Mo et al. demonstrated that KIF2C promotes HCC through the Ras/MAPK and PI3K/AKT pathways [62]. Cyclin-dependent kinase 1 (CDK1), a crucial cell cycle regulator, is elevated in liver, colorectal, and prostate cancers [34,63,64]. GMNN, another cell cycle regulator [65], is overexpressed in liver, colorectal, pancreatic, and breast cancers [66,67,68,69,70]. Aurora kinase A (AURKA), a serine/threonine kinase critical for mitosis, is overexpressed in tumors compared to non-cancerous tissues [71]. However, their strong predictive performance in cancer progression underscores the need for further research into cellular senescence in cancer development.

Our research also highlights two senescence markers, *CD34* and *NOX4*, which exhibit high expression in endothelial cells. CD34, a transmembrane glycoprotein on stem/progenitor cells, serves as a marker for endothelial differentiation and angiogenic tumors, often used to assess vascular invasion [72]. Contrary to previous studies suggesting *CD34* may decline during senescence [73], our findings show robust *CD34* expression in endothelial cells, implicating its role in tumor angiogenesis. This facilitates tumor growth and dissemination by supplying essential nutrients and oxygen, making *CD34* a promising marker for tumor development and metastasis. Similarly, *NOX4*, a NADPH oxidase generating ROS, is closely linked to oxidative stress, a key contributor to cellular senescence [74]. Excessive ROS production under oxidative stress can cause cellular damage and modulate the microenvironment, fostering cancer initiation and progression [75]. Peñuelas-Haro et al. found that the loss of *NOX4* induces metabolic reprogramming via an NRF2/MYC-dependent manner, promoting HCC progression [76]. The heightened expression of *NOX4* in endothelial cells suggests its potential as a marker for tumor development.

### 3.2. Immunological Microenvironment and Therapeutic Strategies

Through our comprehensive analysis of the immunological microenvironment, we revealed that during disease progression, the number and proportion of immune cells, like NK cells and T cells, increase. However, their overall immune function and cytotoxicity decline, creating a more favorable environment for cancer cell survival and promoting disease progression. Our experimental findings support the following hypothesis: after *BIRC5* knockdown, NK cell proliferation increased, but their ability to inhibit HCC cells and their cytotoxicity decreased. These results suggest that HCC cells enhance immune evasion, aligning with our analysis of immune escape mechanisms and highlighting the critical role of immune escape in cancer progression despite higher immune cell numbers. The microenvironment plays a crucial role in cancer development [77]. Thus, our research underscores the importance of the immunological microenvironment in immune cell senescence and the potential for immunotherapy in HCC.

The cellular senescence of tumor-infiltrating immune cells, such as T cells, is a significant aspect of immune dysfunction induced by various malignant tumors [78]. Consistent with our research, evidence indicates that senescent T cells within suppressive tumor microenvironments are potential targets for tumor immunotherapy, emphasizing the need to understand their roles in tumor immunity for developing novel immunotherapeutic strategies [79]. T cell senescence is associated with immune evasion and tumor progression, highlighting the necessity of addressing it in cancer therapy [80]. Our research also indicates that NK cells contribute to cancer progression through cellular senescence. Survivin, encoded by *BIRC5*, is an inhibitor of apoptosis and plays a pivotal role in regulating cell survival and proliferation. However, the increased proliferation and altered cell cycle distribution observed in *BIRC5*-knockdown NK92 cells cannot be fully attributed to reduced apoptosis alone. Our experimental findings indicate that *BIRC5* knockdown affects both apoptosis regulation and critical cell cycle checkpoints. Specifically, it enhances NK92 cell proliferation and promotes cell cycle progression, particularly by facilitating the transition from the G1 to G2 phase (Figure 6B,C), suggesting a direct effect on cell cycle regulation beyond apoptosis inhibition. This is further supported by reduced ROS levels, suggesting a broader impact on cellular stress responses (Figure 6D). Collectively, these results imply that *BIRC5* plays a dual role in both regulating apoptosis and controlling the cell cycle in NK cells, underlying the observed changes in proliferation and immune function.

Considering the low sensitivity of traditional diagnostic techniques and the lack of obvious early symptoms, liver cancer is often diagnosed at an advanced stage [81,82]. Despite recent advances in HCC treatment, many patients still experience treatment resistance and disease progression [83,84]. Our study provides valuable insights for devising immunotherapy strategies. However, we believe that our work could be further explored in aspects such as the potential impact of cohort heterogeneity on the accuracy of machine learning models and the exclusive reliance on transcriptomic data, which limits the scope of our findings. Additionally, while our experimental validation of *BIRC5* in NK92 cells is promising, further functional assays and larger clinical studies are necessary to confirm these results and their clinical relevance, as well as to enhance the interpretability of the machine learning models. However, our current study presents an innovative approach by using machine learning algorithms to identify senescence-related markers in HCC, providing a robust classification model for patient stratification. We validated the prognostic value of these markers, offering a more precise tool for predicting survival and personalizing treatment. Additionally, our analysis of the immune microenvironment reveals the negative prognostic impact of *BIRC5* expression in NK cells, highlighting its role in immune evasion and as a potential therapeutic target in advanced HCC. The classification method we propose aims to improve the prognostic assessment of liver cancer and identify patients who may benefit from immunotherapy. By utilizing HCC-CSM expression levels, clinicians can predict patient responses to immunotherapy, thereby facilitating the development of personalized strategies to improve patient outcomes. Furthermore, this study also aims to develop therapeutic approaches that target NK cells to reduce the expression of the *BIRC5* gene, potentially alleviating the condition of patients with severe HCC.

## 4. Materials and Methods

### 4.1. Data Collection and Preprocessing

We obtained transcriptomic data, clinical data, and meta-information of TCGA-LIHC and ICGC-LIRI-JP from Xena [85] (https://xena.ucsc.edu/, accessed on 24 April 2024, TCGA-LIHC: N = 463, ICGC-LIRI-JP: N = 232). A total of 463 patients with survival data was available in the TCGA-LIHC cohort, with 424 samples included in the RNA sequencing analysis. Tumor samples were identified based on the “Primary Tumor” label, resulting in 371 tumor samples. Normal solid tissue samples were identified based on the “Solid Tissue Normal” label, totaling 50 samples. Additionally, for the ICGC-LIRI-JP cohort, there were 260 donors, of whom 232 had survival data. A total of 437 samples underwent RNA sequencing. Tumor samples were classified based on the “Primary tumor-solid tissue” label, resulting in 197 tumor samples, and 240 normal solid tissue samples were identified based on the “Normal-solid tissue” label. The additional validation cohort for HCC-CSMs comes from GSE214846, which includes 65 HCC tissues and 65 normal controls [86].

For TCGA-LIHC, RNA sequencing was performed using the Illumina mRNA TruSeq Kit (RS-122-2001 or RS-122-2002), which converted 1 μg of total RNA into an mRNA library. The library was sequenced on an Illumina HiSeq 2000 platform with a 48 × 7 × 48 bp read length. FASTQ files were generated by CASAVA. RNA reads were aligned to the hg19 genome assembly using MapSplice 0.7.4 [87]. Gene expression corresponding to transcript models from TCGA gAF2.1 (https://TCGA-data.nci.nih.gov/docs/gaf/gaf.hg19.june2011.bundle/outputs/TCGA.hg19.june2011.gaf) was quantified using RSEM.

For ICGC-LIRI-JP, RNA sequencing was performed according to the Illumina protocol to construct RNA-seq libraries, which were then sequenced on the HiSeq 2000 instrument. Short reads that met the following criteria were collected: (i) average base quality ≥ 30 and (ii) unmapped or having ≤ 30 matching bases. The short reads were then aligned to the hg19 reference genome using BLAT, which includes unplaced genomic sequences (represented by overlap group names appended to standard chromosome names, such as chr1_gl000191_random or by the name “chrUn”, followed by the overlap group identifier, such as chrUn_gl000211). Sequences with more than 50 matching bases were removed.

RNA sequencing of GSE214846 was performed by Hepalos Bio. The raw sequencing reads were preprocessed by fastp v0.23.0 [88], and HISAT2 (hierarchical indexing for spliced alignment of transcripts) [89] was used to align the transcriptome sequencing reads to the reference genome, and HTSeq [90] was used for the reads count calculation.

The bulk transcriptome data used were all generated through pair-end sequencing. The original count matrix was used for DESeq2 differential gene expression analysis, and the normalization method for other transcriptome data was log(TPM + 1). TCGA data were used as the training dataset, while ICGC and GSE214846 were the testing dataset. A set of 866 cell senescence-related genes (the CellAge gene set) was sourced from the Human Ageing Genomic Resources (HAGR) [91].

### 4.2. Feature Selection Method and Validation Based on LASSO Regression

The LASSO linear regression analysis was utilized to compare the prognostic differences between differentially expressed genes and cell senescence-related genes. Genes with non-zero coefficients were selected through ten-fold cross-validation. TCGA-LIHC was employed as the training dataset (N = 371), while ICGC-LIRI was utilized as the testing dataset (N = 232). Subsequently, a gene signature associated with the relevant gene set was established using the following formula:$$RiskScore=\sum Coefficient\;of\;gene\left(i\right)*Expression\;of\;gene(i)$$

The coefficient of gene(i) indicates the regression coefficient of gene i and the expression of gene(i) indicates its expression level. Cases were categorized into low-risk and high-risk groups based on the median RiskScore. Kaplan–Meier curves for overall survival were generated, and TimeROC software was used to produce 1-, 3-, and 5-year ROC curves using the R package “timeROC” (0.4) [92]. Univariate and multivariate Cox regression analyses were performed to evaluate the potential of RiskScore as an independent prognostic indicator.

### 4.3. Establishment and Validation of an HCC Diagnostic Model Based on Machine Learning Algorithms

To discern key feature genes distinguishing HCC from normal tissue, we employed three machine learning algorithms, namely KNN, RF, and SVM by the R packages “e1071” (1.7.14), “kernlab” (0.9.32), and “caret” (6.0.94), alongside a feature selection algorithm [93,94,95]. Variable selection was carried out using 10-fold cross-validation, and the models were assessed using MSE, RMSE, and Rsquared. The common feature genes identified by the three machine learning algorithms were termed HCC-CSMs. Subsequently, we computed the AUC values for the selected feature genes using the ICGC-LIRI-JP (N = 437) and GSE214846 (N = 130) datasets to evaluate their classification performance.

### 4.4. Consensus Clustering for Subtyping and Prognostic Correlation

To explore the interrelationships among senescence genes, we performed consensus clustering based on the gene expression levels of HCC-CSMs from TCGA-LIHC using the R package “ConensusClusterPlus” (1.66.0) [96]. Forest plots were employed to represent the effect sizes and their respective confidence intervals from individual studies, as well as to estimate overall summary effect sizes. The hazard ratio (HR) is commonly used in survival analysis to compare the ratio of death occurrence risk between different groups. A HR value less than 1 indicates a lower risk, while a value greater than 1 indicates a higher risk. Utilizing TCGA-LIHC’s survival and meta-information, we conducted a meta-analysis, examining features such as “Age”, “Sex”, “Grade”, “Pathologic_T” (local invasion; the primary determinant of staging), and Stage extracted from TCGA-LIHC’s meta-information. Subgroup forest plots were generated using the R package “forestploter” (1.1.2) [97].

### 4.5. Differential Expression Gene Analysis and Gene Enrichment Analysis

We utilized the “DESeq2” (1.42.1) [98] to conduct differential gene expression analysis, with a threshold of |log2FoldChange| > 1 and FDR > 0.05. GO and KEGG gene enrichment analysis using the R package “clusterprofiler” (4.10.1) [99], with a threshold of FDR > 0.05 for pathway enrichment selection and sorted based on FDR values, was performed.

### 4.6. Somatic Cell Mutation Analysis

Somatic mutation information corresponding to the TCGA-LIHC cohort was obtained from the TCGA dataset. Tumor mutational burden was defined as the number of somatic, non-synonymous, and coding indel mutations per million bases of the genome, using a 5% detection limit. The “maftools” [100] R package (2.18.0) was utilized to calculate the number of somatic non-synonymous point mutations in each sample.

### 4.7. Immune Infiltration Analysis

We performed immune infiltration analysis on cluster 1 and cluster 2 using the R package “cibersort” (1.03, http://cibersort.stanford.edu, accessed on 24 April 2024), obtaining proportions of different immune cells. To investigate variations in immunotherapeutic responses among different subtypes, the TIDE algorithm and the expression levels of immune checkpoint genes (ICGs) were utilized. The TIDE score was calculated using the official TIDE website (https://TIDE.dfci.harvard.edu/, accessed on 24 April 2024). The set of ICGs was obtained from Salomonis et al. [101].

### 4.8. Single-Cell RNA-Seq Data Processing and Analysis

For single-cell transcriptome data analysis, the dataset GSE149614 from the GEO database (GSM4735102, GSM4735106, GSM4735112, GSM4735116, GSM4735119, GSM4735131, GSM4735136, GSM4735142, GSM4735149, GSM4955425) was accessed, including single-cell RNA sequencing data from 9 primary tumor and 9 non-tumor liver samples. The batch effects were addressed by the “harmony” R package (1.2.0) [102]. The raw data underwent preprocessing with the “Seurat” R package (4.4.0) [103], ensuring accuracy and reliability. Key metrics such as nCount RNA, nFeature RNA, and mitochondrial gene set contamination were assessed to maintain data integrity. We used the annotation information in the original data to label the cell categories. Both the Sange diagram and the heat map are drawn using “ggplot” R package (3.5.1).

### 4.9. Cell Culture

The human NK-92 cell line (ELK Biotechnology., Wuhan, China) was cultured in NK92-specific culture medium (iCell-h0388-001b, iCell Bioscience, Shanghai, China). The HepG2 cell line was obtained from iCell Bioscience Inc. and maintained in a complete medium supplemented with 10% fetal bovine serum (40130ES76, YEASEN, Shanghai, China) and 1% streptomycin–penicillin. Both cell lines were incubated at 37 °C in a humidified atmosphere containing 95% air and 5% CO_2_, with medium changes performed each day. For the co-culture experiment involving NK-92 and HepG2 cells, a 0.4 μm pore size chamber was first placed in a 24-well plate and the membrane was moistened with culture medium. Approximately 200 μL of NK92 + siRNA-NC or NK92 + si-BIRC5 cell suspension was then added to the upper chamber and incubated at 37 °C for 1–4 h to allow for cell attachment. The chamber was carefully positioned upright in the wells of the culture plate, and HepG2 cells were seeded in the lower chamber. An additional culture medium was added to the chamber, and co-culturing was continued for 48 h. Following this period, HepG2 cells from the lower chamber were collected for subsequent experiments.

### 4.10. SiRNA Sequence and Transfection Experimental Methods

NK92 cells were transfected with siRNA targeting BIRC5(si-BIRC5-1 GGCUGUUCCUGAGAAAUAAdTdT, si-BIRC5-2 CGGGCAGAAACAACUGAAAdTdT, si-BIRC5-3 GCACUUCAGACCCACUUAUdTdT), and the negative control siRNA(si-NC GGCAGAAUCCUGUAUGUAAdTdT). The transfection of siRNA was performed using the Lipo2000 transfection reagent (11668019, Invitrogen, Carlsbad, CA, USA) according to the supplier’s protocol.

### 4.11. RNA Isolation and qPCR

Total RNA was extracted from cells according to the manufacturer’s protocol using TRIpure Total RNA Extraction Reagent (EP013, ELK Biotechnology). Subsequently, 1 μg of RNA was reverse transcribed into cDNA using a cDNA synthesis kit (EQ031, ELK Biotechnology). qPCR analysis was performed with SEnTurbo™ SYBR Green PCR SuperMix (EQ001, ELK Biotechnology) on the QuantStudio 6 Flex System (Life Technologies, Gaithersburg, MD, USA) instrument. The relative mRNA expression levels were normalized to ACTIN and calculated using the 2-ΔΔCT method. The qRT-PCR primer sequences were as follows: BIRC5: forward: 5′-GTCCACCGCAAATGCTTCTA-3′ and reverse: 5′-TGCTGTCACCTTCACCGTTC-3′; ACTIN: forward: 5′-TTCATCGTCGTCCCTAGCCT-3′ and reverse: 5′-ATCTCAGGCCGACTCAGATGT-3′.

### 4.12. Cell Proliferation Assay

To evaluate the effect of BIRC5 silencing on NK92 cell proliferation and the effect of co-culture of BIRC5-silenced NK92 cells and HepG2 cells on cell proliferation, we performed a CCK-8 cell proliferation assay. Cells were seeded in 96-well plates and CCK-8 reagent was added at 48 h of culture. The absorbance at 450 nm was measured using a DR-200Bs microplate reader (Diatek, Wuxi, China), with solvent-treated cells as the control group and the blank well set to zero. Relative proliferation rate was calculated using the following formula: relative proliferation rate (%) = [(control group − blank) − (experimental group − blank)]/(control group − blank) × 100%. Each experiment was repeated six times.

### 4.13. Cell Cycle Assay

A total of 4 × 10^5^ harvested cells were incubated in phosphate-buffered saline (GNM20012, GENOM, Hangzhou, China) and then fixed in 75% ethanol at 4 °C for 24 h. After washing three times with cold PBS, cell cycle analysis was performed using Cell Cycle and Apoptosis Analysis Kit (40301ES60, Yeasen) according to the instructions provided by the manufacturer. Subsequently, cell cycle analysis was performed using flow cytometry (Cytoflex, Beckman, Miami, FL, USA). Each group of experiments was repeated at least three times.

### 4.14. Apoptotic Rate Assay

Apoptosis analysis was performed using BD Pharmingen™ FITC Annexin V Apoptosis Detection Kit I (556547, BD Biosciences, San Jose, CA, USA) according to the manufacturer’s instructions. Samples were then analyzed by flow cytometry. Each group of experiments was repeated at least three times.

### 4.15. Detection of ROS

NK-92 cells were incubated with 10 μM DCFH-DA (S0033, Beyotime, Beijing, China) at 37 °C for 30 min to assess intracellular ROS levels. Fluorescence was recorded using a flow cytometer at 488 nm (excitation) and 525 nm (emission).

### 4.16. Transwell Assay

A cell suspension containing 1 × 10^5^ cells/mL was placed in the upper chamber, while 500 μL of DMEM (SH30022, Hyclone, South Logan, UT, USA) supplemented with 10% fetal bovine serum was added to the lower chamber. After incubation for 24 h at 37 °C with 5% CO_2_, the migrated cells were fixed with 4% paraformaldehyde for 20 min and subsequently stained with 0.1% crystal violet for 20 min. The stained cells were then counted under a microscope (IX51, Olympus, Tokyo, Japan). For assessing cell invasion, the same chamber used for cell migration assays was employed but pre-coated with Matrigel (354248, Corning, Corning, NY, USA). All other procedures were conducted as described for the cell migration assay.

### 4.17. Statistical Analysis

Biological analyses in the medical field were conducted using R software version 4.4.3. The mean values and standard deviations were derived from at least three independent studies. Paired comparisons between two groups were evaluated using Student’s t-test, while comparisons involving more than two groups were assessed using a one-way analysis of variance and Tukey’s test. Statistical significance was indicated as * *p* < 0.05, ** *p* < 0.01, and *** *p* < 0.001.

## 5. Conclusions

Collectively, through computational and experimental techniques, we present a comprehensive analysis of HCC-CSMs. This analysis covers the prediction of clinical outcomes for liver cancer using senescence-related genes, the identification of HCC-CSMs, and a correlation analysis of clinical characteristics across different groups. Moreover, at the single-cell level, our investigation into the immunological microenvironment revealed that as the number of immune cells increases, hepatocytes regain a larger share of the cell population, likely influenced by the HCC-CSMs we identified. Additionally, we found that immune evasion intensifies as cancer progresses. In line with this, our in vitro experiments showed that knocking down *BIRC5*, an HCC-CSM involved in cell division, resulted in increased NK cell numbers and proliferation. Still, they reduced their immune functionality and cytotoxicity, potentially aiding cancer cell survival. These findings provide insights into how cellular senescence within the immune microenvironment drives HCC progression.

## Figures and Tables

**Figure 1 ijms-26-00773-f001:**
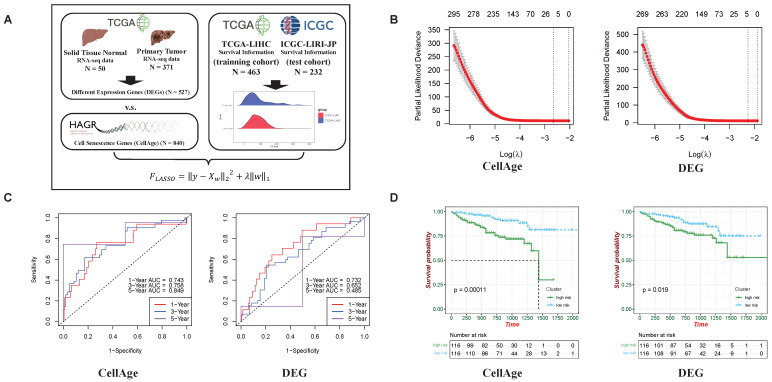
The CellAge gene set predicts the survival of patients with hepatocellular carcinoma (HCC) better than differential expression genes (DEGs). (**A**) We used 463 patient survival information records from the TCGA-LIHC cohort as the training dataset and 232 patient survival information records from the ICGC-LIHC-JP cohort as the testing dataset to evaluate the predictive ability of the CellAge gene set and DEGs for survival using the least absolute shrinkage and selection operator (LASSO) algorithm and Cox regression. DEGs: differential expression analysis between “Solid Tissue Normal” (N = 50) and “Primary Tumor” (N = 371) from TCGA-LIHC; CellAge: obtained from the Human Ageing Genomic Resources (HAGR, https://genomics.senescence.info/, accessed on 13 January 2025). (**B**) Partial likelihood deviance of CellAge and DEGs in the ICGC-LIHC-JP cohort. (**C**) receiver operating characteristic (ROC) analysis of CellAge and DEGs in the ICGC-LIHC-JP cohort. (**D**) Prognosis of high and low risk based on CellAge and DEGs in the ICGC-LIHC-JP cohort.

**Figure 2 ijms-26-00773-f002:**
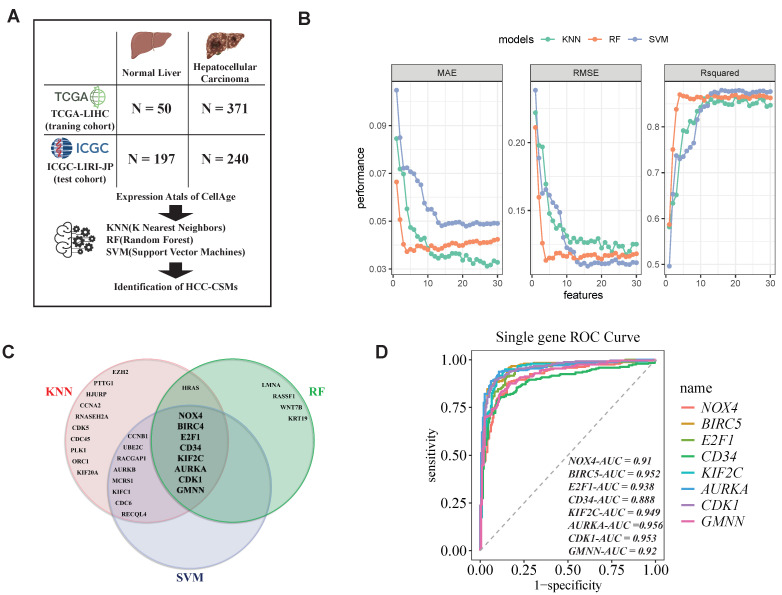
Machine learning identifies markers of HCC. (**A**) We utilized a training dataset comprising 50 normal liver tissue samples from TCGA-LIHC and 371 samples of HCC, along with 197 normal samples and 240 HCC samples from ICGC-LIRI-JP. Employing three machine learning algorithms (KNN, SVM, and RF), we conducted recursive feature elimination to derive the HCC cell senescence markers (HCC-CSMs). (**B**) Evaluation of model performance, including mean squared error (MSE), root mean squared error (RMSE), and Rsquared values, across varying numbers of selected features. (**C**) Venn diagram illustrating the overlap of feature genes identified by different machine learning models. (**D**) ROC curve analysis using the ICGC-LIRI-JP dataset. AUC: area under the curve. KNN: k nearest neighbor. RF: random forest. SVM: support vector machine.

**Figure 3 ijms-26-00773-f003:**
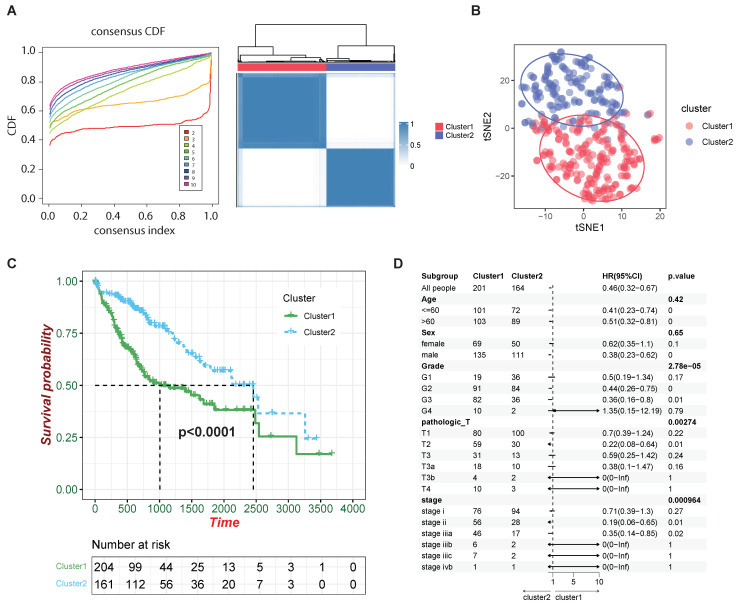
Molecular subtyping and correlation analysis of clinical information. (**A**) Heatmap of the consensus matrix indicating the optimal value for consensus clustering at K =  2. (**B**) t-distributed stochastic neighbor embedding (t-SNE) visualization of gene expression using the CellAge gene set from TCGA-LIHC. (**C**) Survival curve of patients in the two clusters. (**D**) Forest plot generated based on meta-information; non-bold *p*-values indicate the correlation from Cox regression, while bold *p*-values indicate the Fisher correlation within clusters and subgroups.

**Figure 4 ijms-26-00773-f004:**
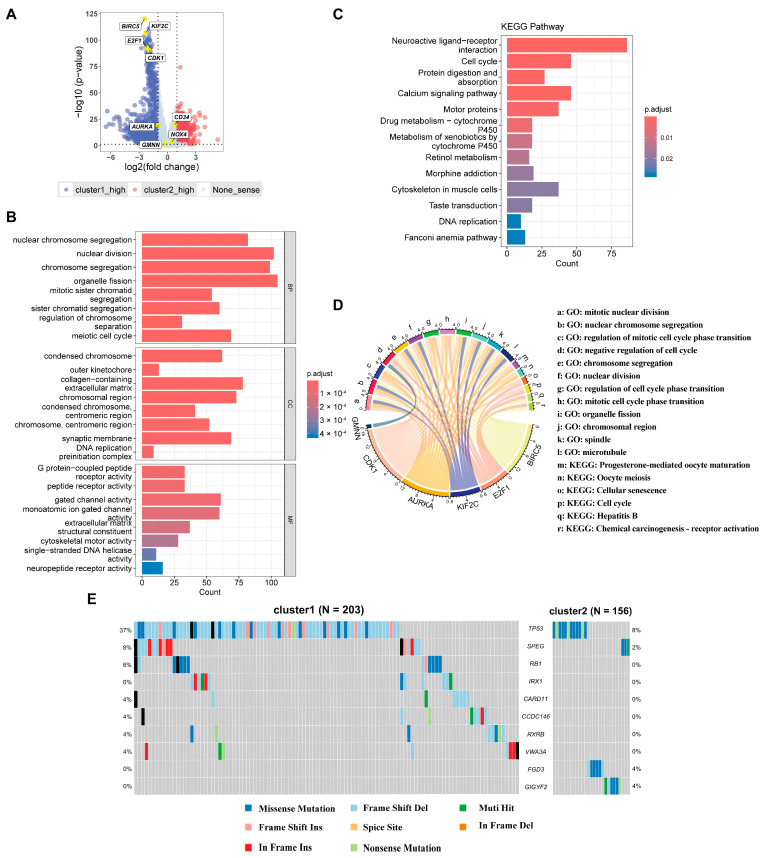
Differentially expressed genes and enrichment analysis. (**A**) Volcano plot showing differentially expressed genes between cluster 1 and cluster 2. (**B**) GO enrichment analysis. (**C**) KEGG enrichment analysis. (**D**) Correspondence between HCC-CSMs and enriched pathways. (**E**) Somatic mutation analysis of cluster 1 and cluster 2.

**Figure 5 ijms-26-00773-f005:**
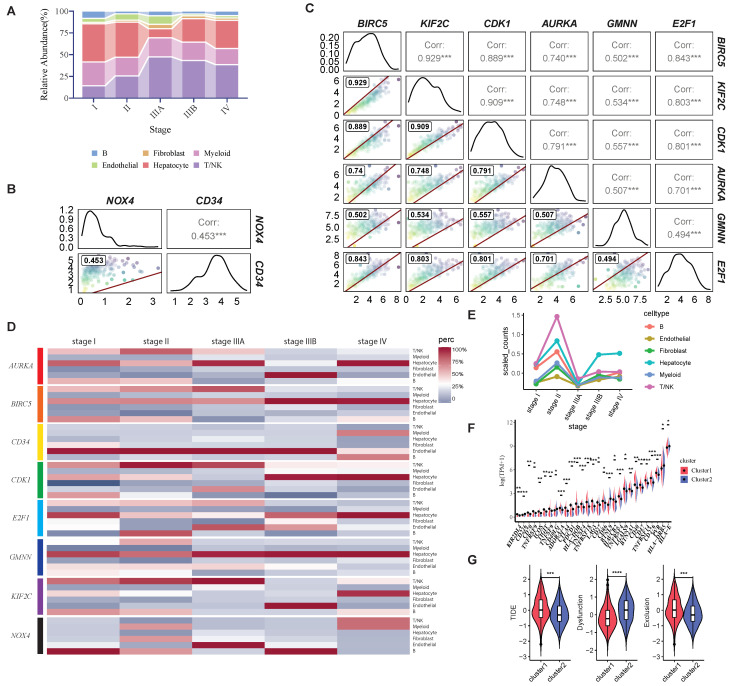
Single-cell gene expression visualization and immune checkpoints, TIDE analysis. (**A**) The proportion of each cell type at different stages in the progression of HCC. (**B**) Correlation between expression levels of *NOX4* and *CD34*. (**C**) Correlation among expression levels of the six genes, *BIRC5*, *KIF2C*, *CDK1*, *AURKA*, *GMNN*, and *E2F1*. (**D**) The proportion of cells expressing HCC-CSMs in different types of cells as the stage changes. (**E**) The average expression of the *BIRC5* gene in different cell types. (**F**) Expression levels of immune checkpoint molecules across different clusters in transcriptome data. (**G**) TIDE scores, dysfunction scores, and exclusion scores across different clusters in transcriptome data. Statistical significance was indicated as * *p* < 0.05, ** *p* < 0.01, and *** *p* < 0.001.

**Figure 6 ijms-26-00773-f006:**
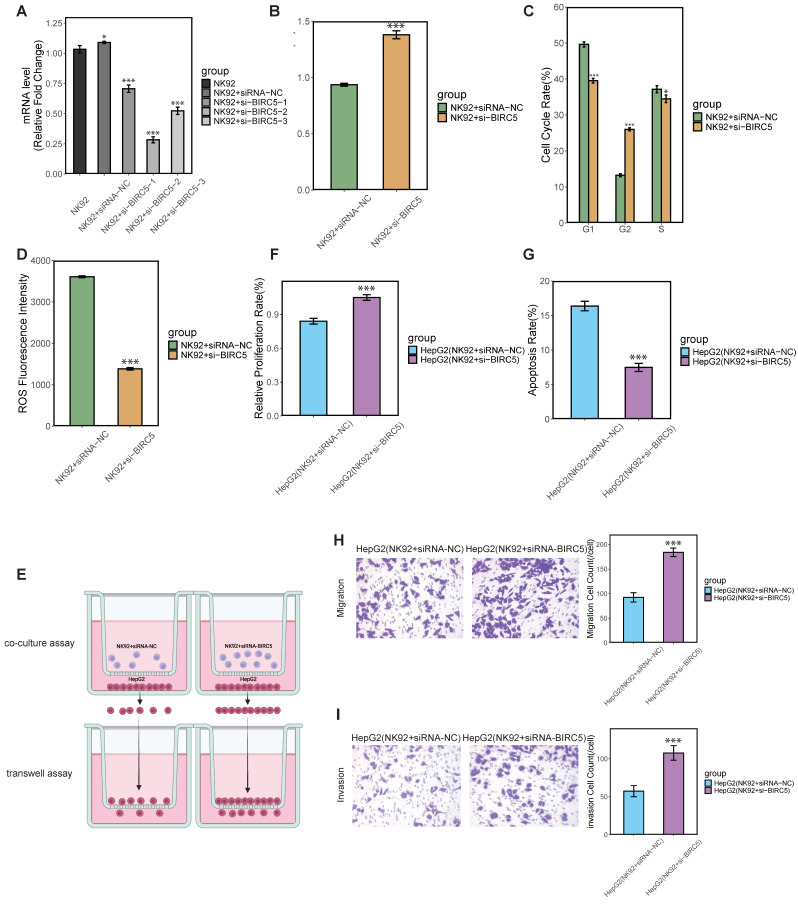
In vitro experiments validate the function of BIRC5 in NK92 cells. (**A**) NK92 cells were transfected with siRNA targeting BIRC5, and cells were collected 48 h later to evaluate BIRC5 knockdown efficiency by quantitative polymerase chain reaction (qPCR). (**B**) NK92 cells transfected with si-BIRC5 were subjected to CCK-8 assay. (**C**,**D**) The cell cycle and reactive oxygen species (ROS) of NK92 cells transfected with si-BIRC5 were detected by flow cytometry. (**E**) Transwell schematic diagram. (**F**,**G**) After NK92 cells transfected with si-BIRC5 were co-cultured with HepG2 cells, CCK-8 cell activity and apoptosis assays were performed on HepG2 cells. (**H**,**I**) Transwell assay for migration and invasion of HepG2 cells co-cultured with NK92 + siRNA-NC, NK92+si-BIRC5. Data are presented as mean ± SD. Statistical significance was indicated as * *p* < 0.05, and *** *p* < 0.001.

## Data Availability

The original data presented in the study are openly available in Xena, Geo database, at [https://xenabrowser.net/datapages/, accessed on 24 April 2024] and [https://www.ncbi.nlm.nih.gov/geo/, accessed on 24 April 2024].

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
