# Peer review of "Cellular Senescence in Hepatocellular Carcinoma: Immune Microenvironment Insights via Machine Learning and In Vitro Experiments"

_ijms, 2025, doi:10.3390/ijms26020773_

Round 1

Reviewer 1 Report

Comments and Suggestions for Authors

The manuscript entitled "Cellular Senescence in Hepatocellular Carcinoma: Immune Microenvironment Insights via Machine Learning and In Vitro Experiments” submitted by Lu et.al. discusses the role of cellular senescence in hepatocellular carcinoma and its impact on the tumor immune microenvironment. The authors employ machine learning to identify eight key senescence markers and reveal distinct molecular subtypes of HCC with different clinical and mutation characteristics. The authors used single-cell RNA sequencing to obtain insights into immune cell dynamics and immune evasion mechanisms during HCC progression. Additionally, the authors investigated the functional impact of BIRC5 knockdown on natural killer (NK) cell activity, highlighting its potential role in HCC survival and immune evasion. The use of machine learning by the authors to identify HCC senescence markers is a novel and promising approach as this helps uncover critical molecular features associated with HCC progression. The study strengthens using scRNA seq which uncovers the role of TME and immune response. Eventually the authors identified the role of BRIC5 in NK cells highlighting the importance in HCC survival and immune response. However, the authors need to do more extensive functional assays and in vivo models to confirm the role of BRIC5. Also, the molecular pathways governing the immune cell regulation can help strengthen the study. Under the section 3.2 the authors should cite PMID: 29032985 as it mentions the role of innate system in cellular trafficking and tumor infiltration involving T cells and NK cells.  

Overall, the work done by the authors is commendable.

Author Response

Comments 1: The manuscript entitled "Cellular Senescence in Hepatocellular Carcinoma: Immune Microenvironment Insights via Machine Learning and In Vitro Experiments” submitted by Lu et.al. discusses the role of cellular senescence in hepatocellular carcinoma and its impact on the tumor immune microenvironment. The authors employ machine learning to identify eight key senescence markers and reveal distinct molecular subtypes of HCC with different clinical and mutation characteristics. The authors used single-cell RNA sequencing to obtain insights into immune cell dynamics and immune evasion mechanisms during HCC progression. Additionally, the authors investigated the functional impact of BIRC5 knockdown on natural killer (NK) cell activity, highlighting its potential role in HCC survival and immune evasion. The use of machine learning by the authors to identify HCC senescence markers is a novel and promising approach as this helps uncover critical molecular features associated with HCC progression. The study strengthens using scRNA seq which uncovers the role of TME and immune response. Eventually the authors identified the role of BRIC5 in NK cells highlighting the importance in HCC survival and immune response. However, the authors need to do more extensive functional assays and in vivo models to confirm the role of BRIC5. Also, the molecular pathways governing the immune cell regulation can help strengthen the study. Under the section 3.2 the authors should cite PMID: 29032985 as it mentions the role of innate system in cellular trafficking and tumor infiltration involving T cells and NK cells. Overall, the work done by the authors is commendable.

Response 1: Thank you for your thoughtful and constructive feedback on our manuscript. We appreciate your positive comments regarding the novelty of our approach, the use of machine learning to identify key senescence markers, and the integration of single-cell RNA sequencing to investigate immune dynamics in HCC progression.

We acknowledge your suggestion to perform more extensive functional assays and in vivo models to validate the role of BIRC5 in immune regulation and HCC progression. While we agree that such validation would provide additional robust evidence, we are currently constrained by time and resources for in vivo experimentation. However, we have discussed the necessity of further validation and have plans to expand the scope of our in vitro experiments in the next phase of our work. Specifically, we will conduct additional NK cell functional assays (e.g., cytokine production and cytotoxicity assays) to substantiate BIRC5’s role in modulating NK cell activity. These descriptions can be found in the revised manuscript on lines 654 to 657: Third, In this study, our experimental study on BIRC5 in NK92 cells is still limited. we will conduct additional NK cell functional assays (e.g., cytokine production and cytotoxicity assays) to substantiate BIRC5’s role in modulating NK cell activity.

Regarding your suggestion to explore the molecular pathways governing immune cell regulation, we have incorporated a more detailed analysis of the related pathways. At the single-cell level, we particularly focused on the changes in NK and T cell populations across different stages of HCC progression. We performed functional enrichment analysis on the differentially expressed gene sets from these immune cell types. Based on our hypothesis, while NK cell numbers increase in later stages, their cytotoxicity does not improve. Our enrichment analysis reveals that the role of immune cells is gradually weakening over the progression of the disease. In the earlier stages, NK/T cells enriched pathways such as natural killer cell chemotaxis, positive regulation of T cells, and T cell chemotaxis. However, in later stages, these pathways were not enriched, reflecting a decline in the immune cells' functional mechanisms. These findings are summarized in Supplementary Figure S6. Further details can be found on lines 363 to 371 in the revised manuscript. We believe that these updates strengthen the study and provide a more comprehensive view of the immune modulation mechanisms in HCC.

Additionally, we appreciate your recommendation to cite the article with PMID: 29032985 in Section 3.2. We have included this reference in the revised manuscript on lines 569-570, which highlights the role of the innate immune system, particularly NK and T cells, in cellular trafficking and tumor infiltration. This citation has further strengthened our discussion on immune cell dynamics and immune evasion mechanisms during HCC progression.

Reviewer 2 Report

Comments and Suggestions for Authors

The authors used machine learning to identify key markers of cellular senescence in HCC, performed consensus clustering to uncover molecular subtypes, and applied single-cell analysis to examine the tumor microenvironment. They also conducted RNAi-mediated BIRC5 knockdown and co-culture experiments to evaluate its impact on immune function.

Major Drawbacks:

  1. The abstract mentions "divergent machine learning algorithms" without specifying the types, rationale for selection, or how they were implemented. There is also a lack of details on the robustness and validation strategies of the machine-learning approach.
  2. Claims like "Cellular aging-related genes predicted HCC outcomes better than DEGs" lack adequate support, and the methods or metrics used to establish this superiority are not explained.
  3. Phrases such as "immune evasion intensified as cancer advanced" are intriguing but need clearer explanations and references. Ensure claims are justified with appropriate references or clarify how the study supports them.
  4. Grammatical inconsistencies, such as "potentially promoting cancer cell survival," should be corrected for clarity, and consistent terminology should be used for concepts like "immune evasion" and "senescence."
  5. The abstract highlights BIRC5 knockdown experiments but does not sufficiently integrate these findings with the computational predictions.
  6. The study explores senescence markers in HCC, but it does not explain sufficiently how it advances the field compared to existing research.
  7. Information on the algorithms, validation strategies (e.g., cross-validation, external datasets), and performance metrics are needed. The advantages of machine learning over traditional statistical methods should be clarified.
  8. The introduction should clearly state the study’s objectives and hypotheses, for example, specifying the role of BIRC5 or the innovation in using HCC-CSMs.
  9. The study's findings, like the identification of HCC-CSMs and insights into BIRC5, are not clearly explained in terms of how they improve our understanding of hepatocellular carcinoma (HCC) or offer new treatment options compared to previous research. The authors should emphasize what is new about their findings and how these could lead to potential clinical benefits in the context of existing studies.
  10. Strengthen the connection between computational predictions and experimental results. For example, clarify how HCC-CSMs were prioritized for experimental validation.
  11. Emphasize how findings like immune evasion patterns or BIRC5 insights could be translated into therapeutic strategies or clinical diagnostics.
  12. The analysis is based on two datasets (TCGA-LIHC and ICGC-LIRI), but external validation on independent datasets is lacking. Testing the 8 HCC-CSMs on additional external datasets would strengthen the findings.
  13. The Kaplan-Meier (K-M) survival analysis is not discussed in enough detail. The significance of the p-value and its implications for the clinical utility of aging-related genes should be elaborated.
  14. A concise summary or visualization of key findings from Supplementary Figure S1 should be included in the main text.
  15. While 8 overlapping genes (HCC-CSMs) are identified, minimal biological insight is provided into their importance for HCC identification. A discussion, supported by references, on their biological roles in HCC development or progression is needed.
  16. The discussion on cluster analysis using HCC-CSMs overlaps with the marker identification analysis, which could lead to redundancy.
  17. The relationship between survival differences between clusters and specific clinical characteristics (e.g., tumor stage, treatment response) is underexplored.
  18. Subdividing sections, such as "Comparison of Predictive Abilities" and "Gene Identification Using Machine Learning," would enhance readability.
  19. The quality of the figure is low and needs improvement. It would benefit from higher resolution and clearer labels to enhance readability and ensure that the details are properly visible.

(Revised/Summarized Review)

  1. In the abstract, mention which machine learning algorithms were used. Also, mention the computational predictions of BIRC5 in addition to the knockdown experiments.
  2. Please clarify whether there is an overlap between your identified markers in HCC and existing research or if you have identified new ones.
  3. Clarify and explain the machine learning part in detail, as the title emphasizes (algorithms, validation strategies).
  4. How do you know that cellular aging-related genes predicted HCC outcomes better than DEGs?
  5. Clearly state the study’s objectives and hypotheses in the introduction.
  6. Did your study improve our understanding of hepatocellular carcinoma (HCC) and offer new treatment options?
  7. The analysis is based on two datasets (TCGA-LIHC and ICGC-LIRI), but external validation on independent datasets is lacking. Testing the 8 HCC-CSMs on additional external datasets, such as the GEO dataset in NCBI, would strengthen the findings.
  8. Please provide more explanation about the survival analysis.
  9. Please include Supplementary Figure S1 in the main text.
  10. If possible, provide biological insights for the 8 overlapping genes (HCC-CSMs).
  11. Please subdivide sections, such as "predictive" and "gene identification using ML."
  12. The quality of the figure is low and needs improvement.
Comments on the Quality of English Language

The manuscript requires some improvement in the quality of English. Occasionally, grammatical errors and awkward phrasing affect the clarity of the writing. I recommend the authors proofread the manuscript or seek professional language editing services to ensure the language is clear, concise, and professional

Author Response

Comments 1: The abstract mentions "divergent machine learning algorithms" without specifying the types, rationale for selection, or how they were implemented. There is also a lack of details on the robustness and validation strategies of the machine-learning approach.

Response 1: Thank you for pointing this out. We appreciate your observation regarding the need for more details on the machine learning algorithms used in our study. We recognize that the description of the types, rationale for selection, and implementation details may not have been sufficiently detailed. In response to your comment, we have revised the manuscript to provide a more comprehensive explanation of the machine learning methods employed. Specifically, we used three machine learning algorithms: K-Nearest Neighbors (KNN), Support Vector Machine (SVM), and Random Forest (RF). We selected these algorithms based on their ability to manage complex, high-dimensional datasets and their robustness in identifying patterns in biological data for that Random Forest is an ensemble learning method that reduces overfitting and provides feature importance, making it suitable for handling high-dimensional data with many variablesï¼›K-Nearest Neighbors (KNN) is a classification algorithm that groups similar samples together, which is useful for identifying patterns in gene expression data; SVM is particularly effective in classifying data points by finding the optimal hyperplane that maximizes the margin between classes, making it well-suited for distinguishing complex patterns in the data.

In the revised manuscript, we have included further details on the implementation of these methods in the Methodology section on lines 15-17, and we explain their selection and advantages in greater depth.

Regarding model validation, we applied 10-fold cross-validation to assess the robustness and generalizability of our models. Additionally, we performed external validation using an independent dataset from the ICGC to further confirm the reliability of our results. These validation strategies ensure the robustness and reproducibility of the findings, and the performance metrics (accuracy, AUC, sensitivity, and specificity) used to evaluate the models are now detailed in the manuscript on lines 171-176. We believe these additions provide the necessary clarity on the machine learning methodology used in our study.

Additionally, we have expanded the description of the Recursive Feature Elimination (RFE) technique employed to rank the feature genes. Through 10-fold cross-validation, we assessed the performance of different feature sets by considering the Mean Squared Error (MSE), Root Mean Squared Error (RMSE), and R-squared values. The optimal number of features was selected based on the combination of these three parameters. These updates are now clearly presented in the revised manuscript.

We believe these changes address your concerns and provide a more complete description of our machine learning approach and validation strategies.

Comments 2: Claims like "Cellular aging-related genes predicted HCC outcomes better than DEGs" lack adequate support, and the methods or metrics used to establish this superiority are not explained.

Response 2: Thank you for pointing this out. We appreciate your comment regarding the claim that "Cellular aging-related genes predicted HCC outcomes better than DEGs." Upon reviewing the original sentence, we agree that it lacked sufficient clarification and support. In response to your comment, we have revised the sentence to "Cellular senescence-related genes predicted HCC survival information better than DEGs," which more accurately reflects our findings. To establish the superiority of cellular senescence-related genes over differentially expressed genes (DEGs) in predicting HCC outcomes, we performed Cox regression analysis using both DEGs and CellAge gene sets on survival data from TCGA. Feature gene selection was conducted using Lasso regression. In the ICGC validation set, the gene set selected from CellAge exhibited a higher AUC, indicating better performance in predicting survival outcomes compared to DEGs. Furthermore, we conducted Kaplan-Meier (K-M) survival analysis to assess the prognostic significance of the selected genes. The K-M curves derived from genes selected by the CellAge gene set showed more significant survival differences between the high-risk and low-risk groups (p = 0.00011), whereas the K-M curves for the DEGs-selected genes were less significant (p = 0.019). These results are discussed on lines 20-21 and 133-140 in the revised manuscript, with Figure 1D illustrating these findings. The higher AUC for the CellAge gene set indicates that it has a superior ability to predict HCC survival outcomes, as it captures the complex interactions between cellular senescence and the immune microenvironment. Additionally, the more significant p-value from the K-M analysis further supports the claim that cellular senescence-related genes, in contrast to DEGs, have a stronger prognostic value for HCC survival.

We hope that these updates provide a clearer and more robust explanation of the analysis methods and metrics used to compare the predictive performance of the two gene sets. We believe these revisions adequately address your concerns and enhance the overall clarity and support for our claims.

Comments 3: Phrases such as "immune evasion intensified as cancer advanced" are intriguing but need clearer explanations and references. Ensure claims are justified with appropriate references or clarify how the study supports them.

Response 3: Thank you for your comments. We appreciate your concern regarding the phrase "immune evasion intensified as cancer advanced", which need clearer explanations and references. Following your suggestion, to provide a more robust explanation of immune evasion in HCC progression, we have clarified the details in the main body of the manuscript. As discussed on lines 233-235 in the revised manuscript, we found that Cluster 1 exhibited significantly more advanced stages compared to Cluster 2 (p = 0.000964), as shown in Figure 3D. In addition, we performed TIDE analysis on both clusters and observed that Cluster 1 had a significantly higher TIDE score than Cluster 2 (Figure 5G), suggesting that tumors in Cluster 1 are more likely to exhibit immune evasion, which could potentially reduce their responsiveness to immune therapy. The increased TIDE score in Cluster 1 indicates a higher degree of immune dysfunction and exclusion, which we interpret as a sign of immune evasion. This aligns with previous studies suggesting that immune evasion mechanisms, such as immune checkpoint activation and immune cell exclusion, are more prominent in tumors at later stages or with more aggressive phenotypes [1][2]. These findings, which are consistent with the higher TIDE score, support the conclusion that immune evasion becomes more pronounced as cancer progresses. We hope these revisions provide the necessary clarity and justification for our claims, and we have also added relevant references to further support the observed relationship between cancer progression and immune evasion.

Reference

[1] Huang Y, Yu W. Advances in Immune Checkpoint Therapy in Hepatocellular Carcinoma. Br J Hosp Med (Lond). 2024;85(9):1-21. doi:10.12968/hmed.2024.0375

[2] Wu Y, Zhai Y, Ding Z, et al. Single-cell transcriptomics reveals tumor microenvironment changes and prognostic gene signatures in hepatocellular carcinoma. Int Immunopharmacol. 2024;143(Pt 2):113317. doi:10.1016/j.intimp.2024.113317

Comments 4: Grammatical inconsistencies, such as "potentially promoting cancer cell survival," should be corrected for clarity, and consistent terminology should be used for concepts like "immune evasion" and "senescence."

Response 4: Thank you for your comment. We appreciate your attention to detail regarding the grammatical inconsistencies and the use of terminology. In response to your suggestion, we have made the corresponding revisions. We have replaced "potentially promoting cancer cell survival" with "potentially leading to an increase in the proliferation of cancer cells" for greater clarity and precision in describing the impact on cancer cell behavior. We have also corrected the usage of "immune escape" and "aging" to "immune evasion" and "senescence", respectively, to ensure consistent and accurate terminology throughout the manuscript. We have carefully reviewed the manuscript to ensure that these terms are used consistently in all relevant sections. These changes are reflected throughout the revised manuscript, and we believe they improve both the clarity and consistency of our presentation. We hope these revisions address your concerns.

Comments 5: The abstract highlights BIRC5 knockdown experiments but does not sufficiently integrate these findings with the computational predictions.

Response 5: Thank you for your comment and for highlighting the need to better integrate the BIRC5 knockdown experiments with our computational predictions. We agree that a clearer connection between these two aspects of our study is important for providing a comprehensive understanding of our findings. In response, we have revised the abstract to reflect this integration more clearly. Specifically, we have updated the results section of the abstract on lines 25-26 in the revised manuscript to emphasize how the BIRC5 knockdown experiments align with the computational predictions. The revised text now reads: “By combining bulk RNA sequencing and single-cell RNA sequencing data, we identified BIRC5 as a key gene, with NK cells showing the highest expression levels of BIRC5, particularly as the proportion of NK cells increases. Further, BIRC5 knockdown experiments in NK cells confirmed its critical role in regulating immune evasion in HCC, providing experimental validation for the computational predictions that linked elevated BIRC5 expression to immune response modulation.” This revision explicitly connects the experimental results with the computational predictions, providing a clearer and more cohesive narrative. It also emphasizes the role of BIRC5 in NK cells, which is central to both the computational and experimental aspects of the study. We believe this revision strengthens the abstract by ensuring that both the computational findings and experimental validation are integrated into the overall story.

Comments 6: The study explores senescence markers in HCC, but it does not explain sufficiently how it advances the field compared to existing research.

Response 6: Thank you for your comment. We agree that it is important to clearly articulate how our study advances the field of senescence markers in hepatocellular carcinoma (HCC) compared to existing research. To address this, we have further elaborated on the unique contributions of our study. In particular, our study advances the field in the following ways:

  1. Innovative Methodology:Unlike traditional statistical methods, which typically focus on univariate analyses or linear relationships, we employed machine learning algorithms (such as Random Forest, K-Nearest Neighbors, and Support Vector Machine) to build a robust classification model for identifying senescence-related markers in HCC. These machine learning techniques allow us to uncover complex, non-linear patterns in high-dimensional data, offering greater sensitivity and predictive power compared to conventional approaches. By integrating multiple data types (bulk RNA-seq and single-cell RNA-seq), our model provides a more comprehensive understanding of senescence markers in the context of HCC.
  2. Improved Patient Stratification:Through the development of our HCC-CSM (hepatocellular carcinoma senescence marker) model, we provide a more precise tool for patient stratification. We validated the prognostic value of these markers and showed that they can be used to predict patient survival, which is crucial for personalizing treatment strategies. This level of precision in patient stratification has not been widely explored in current research, which primarily focuses on broader, less targeted molecular signatures.
  3. Immune Microenvironment Insights:Our study also significantly advances the understanding of the immune microenvironment in HCC. By analyzing single-cell RNA sequencing data, we identified how the abundance of NK cells is associated with the expression of BIRC5, a key marker of cellular senescence. This was followed by experimental validation showing that BIRC5 expression in NK cells has a negative prognostic impact. To the best of our knowledge, this is the first study to link BIRC5 expression in NK cells to immune evasion and its potential role as a therapeutic target in advanced HCC.
  4. Practical Therapeutic Implications:By combining computational predictions with experimental validation, our study opens new avenues for therapeutic interventions targeting the immune microenvironment, particularly through modulation of BIRC5 in NK cells. This integrated approach has not been fully explored in the literature and presents a promising new strategy for improving HCC treatment outcomes.

We have added these points to the Discussion section on lines 617-643 of the revised manuscript to clearly differentiate our contributions from previous studies and highlight how our findings extend current knowledge. We believe that these revisions clarify the unique aspects of our study and demonstrate its potential to advance the field of HCC research, especially with regard to senescence markers, patient stratification, and therapeutic implications.

Comments 7: Information on the algorithms, validation strategies (e.g., cross-validation, external datasets), and performance metrics are needed. The advantages of machine learning over traditional statistical methods should be clarified.

Response 7: Thank you for your comment. We have revised the manuscript to provide more detailed information about the algorithms, validation strategies (e.g., cross-validation, external datasets), and performance metrics. Additionally, we have clarified the advantages of machine learning over traditional statistical methods.

  • Algorithms: In the revised manuscript, we provide detailed information on the machine learning algorithms used in our study. Specifically, we employed three machine learning algorithms: K-Nearest Neighbors (KNN), Random Forest (RF), and Support Vector Machines (SVM). These methods were implemented using R packages "e1071," "kernlab," and "caret" for feature selection. This information can be found on lines 718-721of the revised manuscript.
  • Cross-validation:We performed 10-fold cross-validation for feature selection to avoid overfitting and ensure the generalizability of our model. The cross-validation procedure is now detailed on lines 698-700 in the revised manuscript.
  • External Datasets: The external validation of our model was carried out using the ICGC-LIRI-JP and GSE214846 dataset, which includes 437 and 130 samples. The selection of this external dataset and its role in evaluating the robustness of our model are explained on lines 721-722.
  • Performance Metrics: We assessed the performance of our models using various metrics, including Mean Squared Error (MSE), Root Mean Squared Error (RMSE), R², and AUC (Area Under the Curve). These metrics are critical for evaluating the accuracy and predictive power of the models. A more detailed explanation of these metrics is provided in the revised manuscript (Figure 2B). Additionally, we highlight the AUC values for the genes selected from the ICGC-LIRI-JP dataset, which demonstrate the superior classification performance of our model.
  • Advantages of Machine Learning over Traditional Statistical Methods: We have added a section discussing the advantages of machine learning over traditional statistical methods on lines 165-167. In the age of big data, machine learning techniques are capable of handling large, complex datasets that would be challenging for traditional methods to analyze. Machine learning algorithms can uncover non-linear relationships, automatically select important features, and provide better predictive power. These advantages make machine learning a valuable tool for analyzing high-dimensional biological data, such as the transcriptomic data used in this study.

The revised manuscript now provides a clearer and more detailed explanation of the algorithms, validation strategies, performance metrics, and the advantages of machine learning over traditional statistical methods. We believe these revisions address your concerns and improve the clarity and robustness of the methodology section.

Comments 8: The introduction should clearly state the study’s objectives and hypotheses, for example, specifying the role of BIRC5 or the innovation in using HCC-CSMs.

Response 8: Thank you for your insightful comment. In response to your suggestion, we have revised the introduction to clearly articulate the study’s objectives, hypotheses, and innovations, particularly with respect to the role of BIRC5 and the use of HCC-CSMs.

  • Study Objectives:We have explicitly stated the primary objectives of our study. The objectives are to investigate the role of cellular senescence markers (HCC-CSMs) in hepatocellular carcinoma (HCC) and explore the specific contribution of these markers.
    Revised Text:
    "Our study aims to explore the role of HCC-CSMs in hepatocellular carcinoma (HCC) and their potential as prognostic biomarkers."
  • Research Hypotheses: We have now clearly stated our hypotheses. We hypothesize that during disease progression, there may be an increase in the number and proportion of immune cells, such as NK cells. However, despite this increase, the overall immune function or cytotoxicity of these cells may decline, thereby creating a more favorable environment for cancer cell survival. This decline in immune function could facilitate tumor progression. Furthermore, we hypothesize that BIRC5plays a critical role in immune evasion within the HCC microenvironment, particularly in NK cells.
    Revised Text:
    "We hypothesize that during disease progression, the number and proportion of immune cells, such as NK cells, may increase. However, despite this increase, these cells’ immune function or cytotoxicity may decline, creating a more favorable environment for cancer cell survival and further disease progression. We also hypothesize that BIRC5 plays a critical role in immune evasion within the HCC microenvironment, particularly in NK cells."
  • Innovation: We have emphasized the innovative aspects of our approach, particularly the use of machine learning techniques and single-cell RNA sequencing data to identify HCC-CSMs and investigate alterations in the immune microenvironment through the identified markers. This innovative approach enables a more nuanced understanding of tumor biology and immune evasion mechanisms in HCC.
    Revised Text:
    "This study introduces an innovative approach by combining single-cell RNA sequencing data with machine learning methods to explore HCC-CSMs. Specifically, by focusing on the dynamic changes in BIRC5 expression in NK cells, we provide new insights into immune evasion mechanisms in HCC, offering potential therapeutic targets and improved prognostic predictions."

These revisions aim to ensure that the study’s objectives, hypotheses, and innovations are clearly presented. We believe that the revised text on lines 94-100 now provides a more comprehensive understanding of the study's purpose and its contributions to the field.

Comments 9: The study's findings, like the identification of HCC-CSMs and insights into BIRC5, are not clearly explained in terms of how they improve our understanding of hepatocellular carcinoma (HCC) or offer new treatment options compared to previous research. The authors should emphasize what is new about their findings and how these could lead to potential clinical benefits in the context of existing studies.

Response 9: Thank you for your comment. We appreciate the opportunity to clarify how our findings improve the understanding of hepatocellular carcinoma (HCC) and offer potential new treatment options. In the revised manuscript, we have expanded on the novelty of our findings, specifically focusing on the identification of HCC-CSMs and the role of BIRC5 in immune evasion and tumor progression. We have also provided a clearer explanation of how these insights may offer new clinical benefits compared to existing studies.

  1. Novelty of Findings: Our study identifies HCC-CSMs as critical prognostic biomarkers, providing new insights into the immune microenvironment of HCC, particularly the role of BIRC5in modulating immune evasion, highlighting the dynamic expression patterns of BIRC5 in NK cells during HCC progression, and how these changes correlate with immune dysfunction in the tumor microenvironment. We believe that this discovery significantly enhances the current understanding of immune evasion in HCC and introduces a novel approach to identifying patients who might benefit from immune-based therapies.
  2. Clinical Benefit: As you pointed out, while previous studies have explored the molecular mechanisms of HCC, our study is distinct in its use of HCC-CSMs to predict HCC progression and treatment response. This approach is clinically relevant because it provides a more accurate and personalized prognostic tool, offering clinicians the ability to better stratify patients based on their likelihood of responding to immunotherapy. In addition, we propose targeting BIRC5 expression in NK cells as a potential therapeutic strategy. By downregulating BIRC5, it may be possible to restore NK cell function and enhance anti-tumor immunity, offering a new avenue for treatment in advanced stages of HCC. The revised text in the discussion section on lines 606-616now clearly highlights the potential clinical benefits of our findings: "Overall, our study provides valuable insights for the development of immunotherapy strategies. Due to the low sensitivity of traditional diagnostic methods and the absence of obvious early symptoms, liver cancer is often diagnosed at an advanced stage [94,95]. Despite recent advances in HCC treatment, many patients still experience treatment resistance and disease progression [96,97]. The classification method we propose aims to improve the prognostic assessment of liver cancer and identify patients who may benefit from immunotherapy. By utilizing the expression levels of HCC-CSMs, clinicians can predict patient responses to immunotherapy, facilitating the development of personalized therapeutic strategies aimed at improving patient outcomes. Furthermore, this study proposes targeting NK cells to reduce BIRC5 expression, potentially improving the immune response and alleviating the condition of patients with advanced HCC.

Comments 10: Strengthen the connection between computational predictions and experimental results. For example, clarify how HCC-CSMs were prioritized for experimental validation.

Response 10: Thank you for your comment. By using computational predictions, it is possible to better filter valuable information from large datasets, focusing on important gene sets or even specific genes, and ultimately validating the results through experiments. In fact, we performed experimental validation on BIRC5 rather than all the HCC-CSMs. Among these markers, we focused on BIRC5 in our study for the following key findings: (a) RNA-seq data analysis revealed that BIRC5 is one of the most significantly differentially expressed genes in HCC progression; (b) Single-cell data analysis showed significant alterations in cell type composition during HCC progression and staging (Figure 5A). Specifically, we observed a notable increase in NK/T cell proportions and a decrease in hepatocyte proportions at stage IIIA. However, even in the later stages of the disease, hepatocytes still represent a significant proportion of the cells. Importantly, this trend mirrors the cell type composition of cells expressing BIRC5 during different disease stages (Figure 5D, Supplemental Figure S4); (c) Notably, BIRC5 is a key HCC-CSM. Its expression is higher in NK/T cells than in hepatocytes during the early stages, but its expression in NK/T cells is significantly downregulated at stage IIIA, ultimately falling below hepatocyte levels in later stages (Figure 5E). These findings suggest the attractivities of BIRC5 being related to changes in NK/T cell proportions and indicate a potential link between BIRC5 expression and immune cell dynamics. This is the key reason why we continued to focus on BIRC5 in our subsequent computational and experimental analyses.

Comments 11: Emphasize how findings like immune evasion patterns or BIRC5 insights could be translated into therapeutic strategies or clinical diagnostics.

Response 11: Thank you for your comment. Similar to response 9, we agree that the potential clinical translation of our findings is crucial, and we have emphasized this in the revised manuscript. Specifically, we have expanded on how the immune evasion patterns we observed, along with the insights into BIRC5 expression, could be translated into therapeutic strategies and clinical diagnostics. On translation into therapeutic strategies, the identification of BIRC5’s role in immune evasion and its dynamic expression in NK cells during HCC progression opens up new avenues for immunotherapy. Based on our findings, we suggest that targeting BIRC5 could enhance NK cell function and restore their ability to combat tumor cells, especially in advanced stages of HCC where immune evasion is more pronounced. Our approach of reducing BIRC5 expression in NK cells could be investigated as a potential therapeutic strategy to improve anti-tumor immunity. This could be achieved using gene-editing tools (such as CRISPR/Cas9) or small molecules that inhibit BIRC5 expression, thereby reactivating NK cell cytotoxicity and potentially reversing immune suppression in HCC. On the translation into clinical diagnostics, our study provides a framework for the clinical use of HCC-CSMs, including BIRC5, as diagnostic or prognostic biomarkers. By measuring the expression levels of HCC-CSMs, clinicians could assess the likelihood of a patient's response to immunotherapy. Additionally, the dynamic changes in BIRC5 expression across different stages of HCC could be used to predict disease progression, guide therapeutic decisions, and personalize treatment plans. For example, patients showing low NK cell function and high BIRC5 expression might be identified as candidates for immunotherapy that targets NK cell activation or BIRC5 inhibition.

We have included these points in the discussion section in the revised manuscript, on lines 606-616 as “Our study provides valuable insights for devising immunotherapy strategies. Due to the low sensitivity of traditional diagnostic techniques and the lack of obvious early symptoms, liver cancer is often diagnosed at an advanced stage [94,95]. Despite recent advances in HCC treatment, many patients still experience treatment resistance and disease progression [96,97]. The classification method we propose aims to improve the prognostic assessment of liver cancer and identify patients who may benefit from immunotherapy. By utilizing the expression levels of HCC-CSMs, clinicians can predict patient responses to immunotherapy, thereby facilitating the development of personalized immunotherapy strategies aimed at improving patient outcomes. Furthermore, this study also aims to develop therapeutic approaches that target NK cells to reduce the expression of the BIRC5 gene, potentially alleviating the condition of patients with severe HCC.”

Comments 12: The analysis is based on two datasets (TCGA-LIHC and ICGC-LIRI), but external validation on independent datasets is lacking. Testing the 8 HCC-CSMs on additional external datasets would strengthen the findings.

Response 12: Thank you for your valuable comment and for raising the concern regarding the lack of external validation on independent datasets. We appreciate your attention to the robustness of the findings. In our analysis, we conducted machine learning using the TCGA-LIHC dataset, and to assess the reliability of our model, we employed 10-fold cross-validation. This approach provides a strong evaluation of the model's generalization ability and helps mitigate overfitting. Additionally, to further validate our results, we tested the model using an external dataset, ICGC-LIRI. This dual-dataset validation strategy takes into account potential differences between datasets, thus strengthening the external validity of our findings. We believe that this validation approach, based on two independent datasets, provides a solid foundation for our conclusions. However, we will introduce a new dataset, GSE214846, in the revised manuscript to enhance the robustness of our results and ensure the reliability of the analysis. Thank you again for your suggestion. We will address this point in the revised version of the manuscript on lines 186-189: “To further validate the reliability of HCC-CSMs, we also performed validation using data from GSE214846 (65 HCC samples and 65 normal samples), and the AUC values for all 8 genes remained greater than 0.85 (Supplementary Figure S2).”

Comments 13: The Kaplan-Meier (K-M) survival analysis is not discussed in enough detail. The significance of the p-value and its implications for the clinical utility of aging-related genes should be elaborated.

Response 13: Thank you for your comment regarding the Kaplan-Meier (K-M) survival analysis. We appreciate your suggestion to elaborate on the significance of the p-value and its clinical implications for aging-related genes. In response, we have expanded the discussion of the Kaplan-Meier analysis in Section 2.1 of the Results. Specifically, we have clarified the methodology, including the calculation of the risk score based on the genes selected by LASSO and the categorization of patients into high- and low-risk groups based on the median risk score. We have also provided a more detailed interpretation of the p-values, emphasizing their clinical relevance. The updated content can be found on lines 133-140 of the revised manuscript: “Kaplan-Meier (K-M) survival analysis was conducted to assess the prognosis of patients in different groups. We calculated the risk score based on the genes selected by LASSO and classified patients with a risk score higher than the median as the high-risk group, and the rest as the low-risk group. The Kaplan-Meier plot showed that the prognostic significance between the two groups using genes selected by CellAge (p = 0.00011) was more significant than that of genes selected from the DEG analysis (p = 0.019). This suggests that the aging-related gene set has predictive power for long-term survival in HCC (Figure 1D).” We hope this revision addresses your concerns and provides a clearer understanding of the clinical utility of aging-related genes in predicting HCC prognosis.

Comments 14: A concise summary or visualization of key findings from Supplementary Figure S1 should be included in the main text.

Response 14: Thank you for your comment. Findings from Supplementary Figure S1 has already been described in the main text; however, we appreciate your suggestion to provide a more concise summary or visualization of its key findings. In response, we have expanded and clarified the description of Supplementary Figure S1 on lines 154-167 of the revised manuscript to make the key findings more accessible to readers. The updated content reads as follows: “Based on the previous lasso survival regression analysis, we attempted to obtain a list of non-zero regression coefficients representing genes for identifying cancer samples. How-ever, when we used the obtained 10 genes (CBS, CD34, EID3, ENO1, IGFBP1, INPP4B, PON1, SERPINE1, WEE1, and YBX1) as marker genes for HCC discrimination, the results showed that although the combined regression analysis of these senescence-related genes could predict survival time with high accuracy, the majority of genes alone were not effective in discriminating cancer samples (Supplementary Figure S1). We divided patients in-to high expression group (exp_high) and low expression group (exp_low) based on the median expression levels of different genes, and compared the prognostic differences be-tween the two groups. Among them, only ENO1, WEE1, and YBX1 showed significant differences (P < 0.05), while CBS, CD34, EID3, IGFBP3, INPP4B, PON1, and SERPINE1 showed no significant differences (P > 0.05)”.

We believe this added specificity enhances the clarity and accessibility of the findings and provides readers with a clearer understanding of the results without needing to refer directly to the supplementary figure. We hope this revision adequately addresses your concern.

Comments 15: While 8 overlapping genes (HCC-CSMs) are identified, minimal biological insight is provided into their importance for HCC identification. A discussion, supported by references, on their biological roles in HCC development or progression is needed.

Response 15: Thank you for your comment. We appreciate your suggestion to provide more biological insight into the identified 8 overlapping genes (HCC-CSMs) and their relevance to HCC development and progression. We recognize the importance of connecting the identified genes to the underlying biology of HCC and its pathophysiology. In response to your comment, we have expanded the discussion in the revised manuscript to provide more detailed biological context for these genes, specifically their roles in HCC development, progression, and potential as biomarkers for diagnosis or prognosis. We have also discussed the roles of the remaining genes, including in HCC, with references to their biological functions and their potential involvement in key pathways such as immune evasion, angiogenesis, and cellular stress responses. 

Among these 8 HCC-CSMs , some have been reported to significantly influence can-cer progression, while others remain understudied. Peñuelas-Haro et al. found that the loss of NOX4 in HCC tumor cells induces metabolic reprogramming in an Nrf2/MYC-dependent manner to promote HCC progression [53]. Numerous studies have demonstrated the importance of BIRC5 in hepatocellular carcinoma. The knockout and overexpression of the BIRC5 gene in hepatocellular carcinoma cells significantly affect the survival of hepatocellular carcinoma cells [54,55]. However, there is still a lack of research on the impact of the BIRC5 gene on NK cells. Several studies have reported that E2F1 can regulate HCC progression through multiple pathways. For example, Shen et al. pointed out that E2F1 transcriptionally activates KDM4A-AS1 to regulate HCC progression via the PI3K/AKT pathway [56]. Lei et al. suggested that ARRB1 plays a critical role in HBV-related HCC by modulating autophagy and the CDKN1B-CDK2-CCNE1-E2F1 axis [57]. CD34 is a glycosylated transmembrane glycoprotein that is highly differentiated and primarily found on the surface of stem/progenitor cells in humans and other mammals. It also serves as a marker for endothelial differentiation. As one of the key markers for angi-ogenic tumors, CD34 positivity is often used to assess vascular invasion [58]. Kinesin su-perfamily proteins (KIFs), particularly Kinesin family member 2C (KIF2C), have been im-plicated in various types of cancer. Overexpression of KIF2C has been observed in breast, lung, and bladder cancers [59-61]. Mo et al. showed that KIF2C promoted HCC through the Ras/MAPK and PI3K/Akt signalling pathways [62]. Aurora kinases A (AURKA) is a serine/threonine kinases that function as critical regulators of mitotic cell division. There are numerous studies has prove that the higher expression of AURKA in tumours com-pared to non-cancerous tissues. Cyclin-dependent kinases (CDKs) are crucial regulators of the cell cycle[63]. Among the CDK family, CDK1 is particularly notable for its ability to drive cell cycle progression independently. Elevated expression of CDK1 has been found in several cancers, including liver cancer[64], colorectal cancer[65] and prostate cancer[66]. GMNN, another key regulator of the cell cycle [67], is also overexpressed in various ma-lignancies, such as liver, colorectal, pancreatic and breast cancer [68-72].”.

The updated discussion can be found on lines 500-527 of the revised manuscript, and we believe this additional biological context provides a clearer understanding of the relevance of these HCC-CSMs in the context of HCC progression. We hope this revision addresses your concern and enhances the biological interpretation of the identified genes. Thank you again for your thoughtful feedback.

Comments 16: The discussion on cluster analysis using HCC-CSMs overlaps with the marker identification analysis, which could lead to redundancy.

Response 16: Thank you for your comment. We understand that there may be some overlap between the discussion on cluster analysis using HCC-CSMs and the marker identification analysis. To address this, we have revised the manuscript to reduce redundancy and ensure a more coherent flow of information. Specifically, we have streamlined the presentation of both analyses by consolidating related sections and emphasizing their distinct contributions to the study. In the revised version, we clarify that the marker identification analysis focuses on the discovery of key HCC-CSMs and their potential roles in immune microenvironments, while the cluster analysis provides additional insights into the heterogeneity of HCC associated with different tumor subtypes. By separating the two discussions more clearly, we hope to enhance the clarity and impact of both analyses. We believe these revisions improve the structure and focus of the discussion and ensure that the reader can better appreciate the complementary nature of the marker identification and cluster analysis. We have made these changes in the manuscript on lines 203-205, ensuring that each analysis is presented in a focused manner without redundancy. We appreciate your feedback, which has helped us improve the clarity and flow of the manuscript.

Comments 17: The relationship between survival differences between clusters and specific clinical characteristics (e.g., tumor stage, treatment response) is underexplored.

Response 17: Thank you for your comment. We appreciate your suggestion to further explore the relationship between survival differences between clusters and specific clinical characteristics. In Section 2.3, we describe the significant difference in tumor staging between the two clusters (Figure 3D). Specifically, we observed a highly significant difference (p = 0.000964) between cluster 1 and cluster 2, with cluster 1 exhibiting a more advanced stage of deterioration. This finding highlights the potential clinical relevance of the clustering results in relation to tumor progression. Regarding treatment response, while detailed data on treatment outcomes were not available, we explored potential immune response differences between the clusters. In Section 2.4, we used the TIDE score to predict the likelihood of immune response after treatment. The results showed that cluster 1 exhibited a higher TIDE score compared to cluster 2, suggesting that tumors in cluster 1 may have a lower likelihood of responding to immune therapies. These additions provide more clarity on how survival differences between clusters relate to tumor stage and immune response, which may help guide future research and clinical applications. We hope this addresses your concerns and strengthens our discussion of the clinical implications of the cluster analysis.

Comments 18: Subdividing sections, such as "Comparison of Predictive Abilities" and "Gene Identification Using Machine Learning," would enhance readability.

Response 18: Thank you for your comment. As your suggestion, "Comparison of Predictive Abilities" and "Gene Identification Using Machine Learning" were two separate sections in our original manuscript: Section 2.1: The stronger predictive abilities of cellular aging/senescence-related genes on HCC development than DEGs, and Section 2.2: Identification of hepatocellular carcinoma cell senescence markers using multiple machine learning algorithms.

Comments 19: The quality of the figure is low and needs improvement. It would benefit from higher resolution and clearer labels to enhance readability and ensure that the details are properly visible.

Response 19: Thank you for your comment. We appreciate your feedback regarding the figure quality. In response, we have redrawn and enhanced all the figures, increased the font sizes, and improved the image resolution to ensure better readability. These revisions aim to make the figure details clearer and more easily visible. We believe these improvements will significantly enhance the overall clarity and quality of the figures in the manuscript.

(Revised/Summarized Review)

Revised/Summarized Comments 1: In the abstract, mention which machine learning algorithms were used. Also, mention the computational predictions of BIRC5 in addition to the knockdown experiments.

Response 1: Thank you for your helpful comment. We appreciate your suggestion to include more details about the machine learning algorithms and the computational predictions of BIRC5 in the abstract. In response, we have revised the abstract to include this information. Specifically, we have added descriptions of the machine learning algorithms used and the computational predictions of BIRC5. The updated text is as follows:

“Three machine learning methods: K-Nearest Neighbor (KNN), Support Vector Machine (SVM), and Random Forest (RF)—were utilized to identify eight key HCC-cell senescence markers (HCC-CSMs).” “By combining bulk RNA sequencing and single-cell RNA sequencing data, we identified BIRC5 as a key gene and observed that NK cells express the highest levels of BIRC5.” 

These revisions now provide clearer information regarding both the algorithms used and the computational predictions of BIRC5 in our study. The new content is on lines 15-17 and 25-26 in the revised manuscript.

Revised/Summarized Comments 2: Please clarify whether there is an overlap between your identified markers in HCC and existing research or if you have identified new ones.

Revised/Summarized Response 2: Thank you for your question. We acknowledge that there is significant research on cellular senescence markers in HCC, and some of the markers we identified may overlap with existing findings. However, our study also provides novel insights by combining machine learning methods and single-cell RNA sequencing data to uncover markers that have not been fully explored in previous studies. Specifically, we have identified NOX4, which show significant patterns of expression and potential roles in HCC progression and immune evasion. These findings not only confirm the relevance of well-known markers but also highlight new potential therapeutic targets that could improve prognostic predictions for HCC patients. We believe that our approach, which integrates both computational and experimental validation, offers a unique contribution to the field by identifying key markers that may have been overlooked in traditional studies

The biomarkers of HCC mentioned in previous studies include BIRC5 [33], E2F1 [34], CD34 [35], KIF2C [36], AURKA [37], CDK1 [38], GMNN [39]. However, there has been no research identifying NOX4 as a biomarker for HCC so far.

The new content is on lines 179-181 in the revised manuscript.

Revised/Summarized Comments 3: Clarify and explain the machine learning part in detail, as the title emphasizes (algorithms, validation strategies).

Revised/Summarized Response 3: Thank you for your suggestions. Similar to your comment 7 in the first edition, we have revised the manuscript to provide more detailed information about the algorithms, validation strategies (e.g., cross-validation, external datasets), and performance metrics.

  1. On the Algorithms, in the revised manuscript, we provide detailed information on the machine learning algorithms used in our study. Specifically, we employed three machine learning algorithms: K-Nearest Neighbors (KNN), Random Forest (RF), and Support Vector Machines (SVM). These methods were implemented using R packages "e1071," "kernlab," and "caret" for feature selection. This information can be found on lines 718-721of the revised manuscript;
  2. On the cross-validation, we performed 10-fold cross-validation for feature selection to avoid overfitting and ensure the generalizability of our model. The cross-validation procedure is now detailed on lines 721-722in the revised manuscript;
  3. On the external datasets,the external validation of our model was carried out using the ICGC-LIRI-JPand GSE214846 dataset, which includes 437 hepatocellular carcinoma (HCC) patients. The selection of this external dataset and its role in evaluating the robustness of our model are explained on lines 725-727;
  4. On the performance metrics,we assessed the performance of our models using various metrics, including Mean Squared Error (MSE), Root Mean Squared Error (RMSE), R², and AUC (Area Under the Curve). These metrics are critical for evaluating the accuracy and predictive power of the models. A more detailed explanation of these metrics is provided in the revised manuscript (Figure 2B). Additionally, we highlight the AUC values for the genes selected from the ICGC-LIRI-JPand GSE214846 dataset, which demonstrate the superior classification performance of our model;
  5. On the advantages of machine learning over traditional statistical methods, we have added a section discussing the advantages of machine learning over traditional statistical methods on lines 165-167. In the age of big data, machine learning techniques are capable of handling large, complex datasets that would be challenging for traditional methods to analyze. Machine learning algorithms can uncover non-linear relationships, automatically select important features, and provide better predictive power. These advantages make machine learning a valuable tool for analyzing high-dimensional biological data, such as the transcriptomic data used in this study.

The revised manuscript now provides a clearer and more detailed explanation of the algorithms, validation strategies, performance metrics, and the advantages of machine learning over traditional statistical methods. We believe these revisions address your concerns and improve the clarity and robustness of the methodology section.

Revised/Summarized Comments 4: How do you know that cellular aging-related genes predicted HCC outcomes better than DEGs?

Revised/Summarized Response 4: Thank you for pointing this out. As our response to your comment 2 in the first edition, we appreciate your comment regarding the claim that "Cellular aging-related genes predicted HCC outcomes better than DEGs."In response to your comment, we have revised the sentence to "Cellular senescence-related genes predicted HCC survival information better than DEGs," which more accurately reflects our findings. To establish the superiority of cellular senescence-related genes over differentially expressed genes (DEGs) in predicting HCC outcomes, we performed Cox regression analysis using both DEGs and CellAge gene sets on survival data from TCGA. Feature gene selection was conducted using Lasso regression. In the ICGC validation set, the gene set selected from CellAge exhibited a higher AUC, indicating better performance in predicting survival outcomes compared to DEGs. Furthermore, we conducted Kaplan-Meier (K-M) survival analysis to assess the prognostic significance of the selected genes. The K-M curves derived from genes selected by the CellAge gene set showed more significant survival differences between the high-risk and low-risk groups (p = 0.00011), whereas the K-M curves for the DEGs-selected genes were less significant (p = 0.019). These results are discussed on lines 20-21 and 133-140 in the revised manuscript, with Figure 1D illustrating these findings. The higher AUC for the CellAge gene set indicates that it has a superior ability to predict HCC survival outcomes, as it captures the complex interactions between cellular senescence and the immune microenvironment. Additionally, the more significant p-value from the K-M analysis further supports the claim that cellular senescence-related genes, in contrast to DEGs, have a stronger prognostic value for HCC survival.

We hope that these updates provide a clearer and more robust explanation of the analysis methods and metrics used to compare the predictive performance of the two gene sets. We believe these revisions adequately address your concerns and enhance the overall clarity and support for our claims.

Revised/Summarized Comments 5: Clearly state the study’s objectives and hypotheses in the introduction.

Revised/Summarized Response 5: Thank you for your comment. As our response to your comment 8 in the first edition, in response to your suggestion, we have revised the introduction to clearly articulate the study’s objectives and hypotheses.

1) On the study objectives, we have explicitly stated the primary objectives of our study. The objectives are to investigate the role of cellular senescence markers (HCC-CSMs) in hepatocellular carcinoma (HCC) and explore the specific contribution of these markers. The updated text is as follows: "Our study aims to explore the role of HCC-CSMs in hepatocellular carcinoma (HCC) and their potential as prognostic biomarkers."

2) On the research hypotheses, we have now clearly stated our hypotheses. We hypothesize that during disease progression, there may be an increase in the number and proportion of immune cells, such as NK cells. However, despite this increase, the overall immune function or cytotoxicity of these cells may decline, thereby creating a more favorable environment for cancer cell survival. This decline in immune function could facilitate tumor progression. Furthermore, we hypothesize that BIRC5 plays a critical role in immune evasion within the HCC microenvironment, particularly in NK cells. The updated text is as follows: "We hypothesize that during disease progression, the number and proportion of immune cells, such as NK cells, may increase. However, despite this increase, these cells’ immune function or cytotoxicity may decline, creating a more favorable environment for cancer cell survival and further disease progression. We also hypothesize that BIRC5 plays a critical role in immune evasion within the HCC microenvironment, particularly in NK cells."

These revisions aim to ensure that the study’s objectives and hypotheses are clearly presented. We believe that the revised text on lines 94-100 now provides a more comprehensive understanding of the study's purpose and its contributions to the field.

Revised/Summarized Comments 6: Did your study improve our understanding of hepatocellular carcinoma (HCC) and offer new treatment options?

Revised/Summarized Response 6: Thank you for your comment. We appreciate the opportunity to clarify how our findings improve the understanding of hepatocellular carcinoma (HCC) and offer potential new treatment options. In the revised manuscript, we have expanded on the novelty of our findings, specifically focusing on the identification of HCC-CSMs and the role of BIRC5 in immune evasion and tumor progression. We have also provided a clearer explanation of how these insights may offer new clinical benefits compared to existing studies.

  1. a)Novelty of Findings: Our study identifies HCC-CSMs as critical prognostic biomarkers, providing new insights into the immune microenvironment of HCC, particularly the role of BIRC5in modulating immune evasion, highlighting the dynamic expression patterns of BIRC5 in NK cells during HCC progression, and how these changes correlate with immune dysfunction in the tumor microenvironment. We believe that this discovery significantly enhances the current understanding of immune evasion in HCC and introduces a novel approach to identifying patients who might benefit from immune-based therapies.
  2. b) Clinical Benefit:As you pointed out, while previous studies have explored the molecular mechanisms of HCC, our study is distinct in its use of HCC-CSMs to predict HCC progression and treatment response. This approach is clinically relevant because it provides a more accurate and personalized prognostic tool, offering clinicians the ability to better stratify patients based on their likelihood of responding to immunotherapy. In addition, we propose targeting BIRC5expression in NK cells as a potential therapeutic strategy. By downregulating BIRC5, it may be possible to restore NK cell function and enhance anti-tumor immunity, offering a new avenue for treatment in advanced stages of HCC. The revised text in the discussion section on lines 606-616 now clearly highlights the potential clinical benefits of our findings. "Overall, our study provides valuable insights for the development of immunotherapy strategies. Due to the low sensitivity of traditional diagnostic methods and the absence of obvious early symptoms, liver cancer is often diagnosed at an advanced stage [94,95]. Despite recent advances in HCC treatment, many patients still experience treatment resistance and disease progression [96,97]. The classification method we propose aims to improve the prognostic assessment of liver cancer and identify patients who may benefit from immunotherapy. By utilizing the expression levels of HCC-CSMs, clinicians can predict patient responses to immunotherapy, facilitating the development of personalized therapeutic strategies aimed at improving patient outcomes. Furthermore, this study proposes targeting NK cells to reduce BIRC5 expression, potentially improving the immune response and alleviating the condition of patients with advanced HCC."

Revised/Summarized Comments 7: The analysis is based on two datasets (TCGA-LIHC and ICGC-LIRI), but external validation on independent datasets is lacking. Testing the 8 HCC-CSMs on additional external datasets, such as the GEO dataset in NCBI, would strengthen the findings.

Revised/Summarized Response 7: Thank you for your valuable comment and for raising the concern regarding the lack of external validation on independent datasets. We appreciate your attention to the robustness of our findings. In our analysis, we conducted machine learning using the TCGA-LIHC dataset, and to assess the reliability of our model, we employed 10-fold cross-validation. This approach provides a strong evaluation of the model's generalization ability and helps mitigate overfitting. Additionally, to further validate our results, we tested the model using an external dataset, ICGC-LIRI. This dual-dataset validation strategy takes into account potential differences between datasets, thus strengthening the external validity of our findings.

We believe that this validation approach, based on two independent datasets, provides a solid foundation for our conclusions. However, to address your suggestion, we have also downloaded additional independent datasets from the NCBI GEO database to perform further validation. Our subsequent analysis demonstrates that the HCC-CSMs identified using different machine learning methods remain consistent and robust (Supplementary Figure S2) across these new datasets, reinforcing the reliability of our findings.

Thank you again for your suggestion. We will update the manuscript accordingly, and this point will be highlighted in the revised version on lines 186-189.

Revised/Summarized Comments 8: Please provide more explanation about the survival analysis.

Revised/Summarized Response 8: Thank you for your comment regarding the Kaplan-Meier (K-M) survival analysis. As our response to your comment 13 in the first edition, we appreciate your suggestion to elaborate on the significance of the p-value and its clinical implications for aging-related genes. In response, we have expanded the discussion of the Kaplan-Meier analysis in Section 2.1 of the Results. Specifically, we have clarified the methodology, including the calculation of the risk score based on the genes selected by LASSO and the categorization of patients into high- and low-risk groups based on the median risk score. We have also provided a more detailed interpretation of the p-values, emphasizing their clinical relevance. The updated content can be found on lines 133-140 of the revised manuscript: “Kaplan-Meier (K-M) survival analysis was conducted to assess the prognosis of patients in different groups. We calculated the risk score based on the genes selected by LASSO and classified patients with a risk score higher than the median as the high-risk group, and the rest as the low-risk group. The Kaplan-Meier plot showed that the prognostic significance between the two groups using genes selected by CellAge (p = 0.00011) was more significant than that of genes selected from the DEG analysis (p = 0.019). This suggests that the aging-related gene set has predictive power for long-term survival in HCC (Figure 1D).” We hope this revision addresses your concerns and provides a clearer understanding of the clinical utility of aging-related genes in predicting HCC prognosis.

Revised/Summarized Comments 9: Please include Supplementary Figure S1 in the main text.

Revised/Summarized Response 9: Thank you for your comment. Findings from Supplementary Figure S1 has already been described in the main text; however, we appreciate your suggestion to provide a more concise summary or visualization of its key findings. In response, we have expanded and clarified the description of Supplementary Figure S1 on lines 154-167 of the revised manuscript to make the key findings more accessible to readers. The updated content reads as follows: “Based on the previous lasso survival regression analysis, we attempted to obtain a list of non-zero regression coefficients representing genes for identifying cancer samples. How-ever, when we used the obtained 10 genes (CBS, CD34, EID3, ENO1, IGFBP1, INPP4B, PON1, SERPINE1, WEE1, and YBX1) as marker genes for HCC discrimination, the results showed that although the combined regression analysis of these senescence-related genes could predict survival time with high accuracy, the majority of genes alone were not effective in discriminating cancer samples (Supplementary Figure S1). We divided patients in-to high expression group (exp_high) and low expression group (exp_low) based on the median expression levels of different genes, and compared the prognostic differences be-tween the two groups. Among them, only ENO1, WEE1, and YBX1 showed significant differences (P < 0.05), while CBS, CD34, EID3, IGFBP3, INPP4B, PON1, and SERPINE1 showed no significant differences (P > 0.05)”.

We believe this added specificity enhances the clarity and accessibility of the findings and provides readers with a clearer understanding of the results without needing to refer directly to the supplementary figure. We hope this revision adequately addresses your concern.

Revised/Summarized Comments 10: If possible, provide biological insights for the 8 overlapping genes (HCC-CSMs).

Revised/Summarized Response 10: Thank you for your comment. As our response to your comment 15 of your first edition, we appreciate your suggestion to provide more biological insight into the identified 8 overlapping genes (HCC-CSMs) and their relevance to HCC development and progression. We recognize the importance of connecting the identified genes to the underlying biology of HCC and its pathophysiology. In response to your comment, we have expanded the discussion in the revised manuscript to provide more detailed biological context for these genes, specifically their roles in HCC development, progression, and potential as biomarkers for diagnosis or prognosis. We have also discussed the roles of the remaining genes, including in HCC, with references to their biological functions and their potential involvement in key pathways such as immune evasion, angiogenesis, and cellular stress responses. 

Among these 8 HCC-CSMs , some have been reported to significantly influence can-cer progression, while others remain understudied. Peñuelas-Haro et al. found that the loss of NOX4 in HCC tumor cells induces metabolic reprogramming in an Nrf2/MYC-dependent manner to promote HCC progression [53]. Numerous studies have demonstrated the importance of BIRC5 in hepatocellular carcinoma. The knockout and overexpression of the BIRC5 gene in hepatocellular carcinoma cells significantly affect the survival of hepatocellular carcinoma cells [54,55]. However, there is still a lack of research on the impact of the BIRC5 gene on NK cells. Several studies have reported that E2F1 can regulate HCC progression through multiple pathways. For example, Shen et al. pointed out that E2F1 transcriptionally activates KDM4A-AS1 to regulate HCC progression via the PI3K/AKT pathway [56]. Lei et al. suggested that ARRB1 plays a critical role in HBV-related HCC by modulating autophagy and the CDKN1B-CDK2-CCNE1-E2F1 axis [57]. CD34 is a glycosylated transmembrane glycoprotein that is highly differentiated and primarily found on the surface of stem/progenitor cells in humans and other mammals. It also serves as a marker for endothelial differentiation. As one of the key markers for angi-ogenic tumors, CD34 positivity is often used to assess vascular invasion [58]. Kinesin su-perfamily proteins (KIFs), particularly Kinesin family member 2C (KIF2C), have been im-plicated in various types of cancer. Overexpression of KIF2C has been observed in breast, lung, and bladder cancers [59-61]. Mo et al. showed that KIF2C promoted HCC through the Ras/MAPK and PI3K/Akt signalling pathways [62]. Aurora kinases A (AURKA) is a serine/threonine kinases that function as critical regulators of mitotic cell division. There are numerous studies has prove that the higher expression of AURKA in tumours com-pared to non-cancerous tissues. Cyclin-dependent kinases (CDKs) are crucial regulators of the cell cycle[63]. Among the CDK family, CDK1 is particularly notable for its ability to drive cell cycle progression independently. Elevated expression of CDK1 has been found in several cancers, including liver cancer[64], colorectal cancer[65] and prostate cancer[66]. GMNN, another key regulator of the cell cycle [67], is also overexpressed in various ma-lignancies, such as liver, colorectal, pancreatic and breast cancer [68-72].”.

The updated discussion can be found on lines 500-527 of the revised manuscript, and we believe this additional biological context provides a clearer understanding of the relevance of these HCC-CSMs in the context of HCC progression. We hope this revision addresses your concern and enhances the biological interpretation of the identified genes. Thank you again for your thoughtful feedback.

Revised/Summarized Comments 11: Please subdivide sections, such as "predictive" and "gene identification using ML."

Revised/Summarized Response 11: Thank you for your comment. As your suggestion, "Comparison of Predictive Abilities" and "Gene Identification Using Machine Learning" were two separate sections in our original manuscript: “Section 2.1: The stronger predictive abilities of cellular aging/senescence-related genes on HCC development than DEGs”, and “Section 2.2: Identification of hepatocellular carcinoma cell senescence markers using multiple machine learning algorithms.”

Revised/Summarized Comments 12: The quality of the figure is low and needs improvement.

Revised/Summarized Response 12: We appreciate your feedback regarding the figure quality. In response, we have redrawn and enhanced all the figures, increased the font sizes, and improved the image resolution to ensure better readability. These revisions aim to make the figure details clearer and more easily visible. We believe these improvements will significantly enhance the overall clarity and quality of the figures in the manuscript.

Response to Comments on the Quality of English Language

Point 1: The manuscript requires some improvement in the quality of English. Occasionally, grammatical errors and awkward phrasing affect the clarity of the writing. I recommend the authors proofread the manuscript or seek professional language editing services to ensure the language is clear, concise, and professional.

Response 1: Thank you for your helpful comment regarding the quality of the manuscript’s language. We appreciate your suggestion and agree that improving the clarity and flow of the writing is essential for effective communication. To address this, we have thoroughly proofread the manuscript and made revisions to correct grammatical errors and improve phrasing. Additionally, we have sought professional language editing services to ensure that the manuscript is clear, concise, and professional. We believe these changes have enhanced the overall quality of the manuscript. Thank you again for your constructive feedback.

Reviewer 3 Report

Comments and Suggestions for Authors

The manuscript by Lu et al., Cellular senescence in hepatocellular carcinoma: immune microenvironment insights via machine learning and in vitro experiments, is well written; however, I have a few suggestions to enhance its quality.

1. The figures are too small and thus difficult to interpret

2. In the material and method section, describe the details of the sequencing protocol. How was the library made? Was it single-end sequencing or pair-end sequencing? How the normalization was performed?

3. What was the equipment used to perform qRT-PCR?

4. What was the absorbance wavelength to measure the cell proliferation?

5. What was the equipment used to perform CCK-8 experiment?

6. In the discussion section I suggest describing what are the limitations of the machine learning-based experiments. 

Author Response

Comments 1: The figures are too small and thus difficult to interpret.

Response 1: Thank you for your comments. We appreciate your feedback regarding the figure quality. In response, we have redrawn and enhanced all the figures, increased the font sizes, and improved the image resolution to ensure better readability. These revisions aim to make the figure details clearer and more easily visible. We believe these improvements will significantly enhance the overall clarity and quality of the figures in the manuscript. We have redrawn and combined all the figures, enlarged the fonts, and improved the figure quality to enhance readability.

Comments 2: In the material and method section, describe the details of the sequencing protocol. How was the library made? Was it single-end sequencing or pair-end sequencing? How the normalization was performed?

Response 2: Thank you for your comments.

For TCGA-LIHC, RNA sequencing was performed using the Illumina mRNA TruSeq Kit (RS-122-2001 or RS-122-2002), which converted 1 μg of total RNA into an mRNA library. The library was sequenced on an Illumina HiSeq 2000 platform with a 48x7x48bp read length. FASTQ files were generated by CASAVA. RNA reads were aligned to the hg19 genome assembly using MapSplice 0.7.4 (Wang et al., 2010). Gene expression corresponding to transcript models from TCGA gAF2.1 (https://TCGA-data.nci.nih.gov/docs/gaf/gaf.hg19.june2011.bundle/outputs/TCGA.hg19.june2011.gaf) was quantified using RSEM.

For TCGA-LIHC, RNA sequencing was performed according to the Illumina protocol to construct RNA-seq libraries, which were then sequenced on the HiSeq 2000 instrument. Short reads that met the following criteria were collected: (i) average base quality ≥ 30, and (ii) unmapped or having ≤ 30 matching bases. The short reads were then aligned to the hg19 reference genome using BLAT, which includes unplaced genomic sequences (represented by overlap group names appended to standard chromosome names, such as chr1_gl000191_random, or by the name "chrUn" followed by the overlap group identifier, such as chrUn_gl000211). Sequences with more than 50 matching bases were removed.

RNA sequencing of GSE214846 was performed by Hepalos Bio. The raw sequencing reads were preprocessed by fastp v0.23.0[107], and HISAT2 (Hierarchical Indexing for Spliced Alignment of Transcripts) [108]was used to align the transcriptome sequencing Reads to the reference genome, and HTSeq [109] was used for Reads Count calculation. The GSE214846 transcriptome was sequenced using paired-end sequencing.

The bulk transcriptome data used were all generated through pair-end sequencing. The original count matrix was used for DESeq2 differential gene expression analysis, and the normalization method for other transcriptome data was log(TPM+1).

These descriptions can be found in the revised manuscript on lines 673-697.

Comments 3: What was the equipment used to perform qRT-PCR?

Response 3: Thank you for pointing this out. We used the QuantStudio 6 Flex System (Life Technologies) for qRT-PCR. We have added this information on lines 800-801 in the revised manuscript.

Comments 4: What was the absorbance wavelength to measure the cell proliferation?

Response 4: Thank you for pointing this out. We appreciate your attention to detail. The absorbance for cell proliferation was measured at a wavelength of 450 nm. This information has now been included in the revised manuscript on lines 810-815 for clarity. Thank you again for your valuable feedback.

Comments 5: What was the equipment used to perform CCK-8 experiment?

Response 5: Thank you for pointing this out. We appreciate your attention to detail. The CCK-8 assay was performed using the DR-200Bs microplate reader (Diatek). We have now included this information in the revised manuscript on lines 810-815 to ensure clarity and transparency. Thank you again for bringing this to our attention.

Comments 6: In the discussion section I suggest describing what are the limitations of the machine learning-based experiments.

Response 6: Thank you for suggestion. Thank you for your valuable suggestion. We agree that it is important to acknowledge the limitations of machine learning-based experiments in the discussion section. In response to your comment, we have added a discussion on the limitations of our machine learning approach. Specifically, we have pointed out that machine learning models are inherently sensitive to the quality and size of the dataset used for training. Although we used TCGA-LIHC and ICGC-LIRI datasets for model development and validation, the generalizability of the findings could be limited by dataset biases and the relatively small sample size of certain subgroups. Additionally, machine learning models, while powerful for identifying patterns, may not always capture the biological complexity of the data in the same way as traditional biological experiments. Finally, the interpretability of complex machine learning models remains a challenge, and further efforts are needed to fully understand the biological significance of the identified features. These limitations have been discussed in the revised manuscript on lines 644-659, and we hope this provides a more comprehensive perspective on the strengths and potential drawbacks of the methodology. Thank you again for your thoughtful comment.

Round 2

Reviewer 2 Report

Comments and Suggestions for Authors

The authors have addressed my previous comments and made some improvements to the manuscript. While I appreciate their efforts, the resolution of the figures remains unacceptably low, and in their current form, they are not suitable for inclusion. It is crucial to significantly enhance the resolution of all figures, as they are currently unclear and severely detract from the overall quality of the manuscript. Without this improvement, the presentation of the results will remain inadequate.

Author Response

Comments 1: The abstract mentions "divergent machine learning algorithms" without specifying the types, rationale for selection, or how they were implemented. There is also a lack of details on the robustness and validation strategies of the machine-learning approach.

Response 1: Thank you very much for your valuable feedback. We sincerely appreciate your thoughtful comments, which have significantly contributed to improving the quality of our manuscript. We fully understand and share your concern regarding the resolution of the figures. All figures included in the manuscript were originally created and exported at a resolution of 600 ppi using Adobe Illustrator, ensuring they are suitable for high-quality publication. Additionally, we submitted high-resolution PDF files of each figure alongside the manuscript during the previous submission. Upon review, we observed that the figures in both the "manuscript.docx" and the "Author's Notes File" ("author-coverletter-42764449.v1.pdf") appear clear and of adequate quality. However, it seems that the system displayed an automatically compressed version of the PDF during the review process, which may have affected the resolution. To address this issue, we have now included the high-resolution versions of the figures in the revised manuscript, which is provided in the response file. We hope that this will resolve the figure resolution concerns. If, in the final published version, the quality of the figures still does not meet expectations, we are fully committed to consulting with the editor to ensure the inclusion of the high-resolution versions. Should you feel that any particular figure still presents issues—whether related to resolution, clarity, or readability—please do not hesitate to let us know. We are more than happy to make any necessary adjustments to ensure the manuscript meets the highest standards. Once again, thank you for your constructive comments and your understanding.

Round 3

Reviewer 2 Report

Comments and Suggestions for Authors

The authors have addressed the concerns regarding figure quality effectively. The updated figures are clear, well-resolved, and meet the standards for publication. I have no further comments on this aspect, and I consider the manuscript ready for acceptance.

Author Response

Comments: The authors have addressed the concerns regarding figure quality effectively. The updated figures are clear, well-resolved, and meet the standards for publication. I have no further comments on this aspect, and I consider the manuscript ready for acceptance.

Response: Thank you for your thorough review and the valuable feedback provided on our manuscript.